# The Male Mouse Meiotic Cilium Emanates from the Mother Centriole at Zygotene Prior to Centrosome Duplication

**DOI:** 10.3390/cells12010142

**Published:** 2022-12-29

**Authors:** Pablo López-Jiménez, Sara Pérez-Martín, Inés Hidalgo, Francesc R. García-Gonzalo, Jesús Page, Rocio Gómez

**Affiliations:** 1Unidad de Biología Celular, Departamento de Biología, Universidad Autónoma de Madrid, 28049 Madrid, Spain; 2Departamento de Bioquímica, Facultad de Medicina, Universidad Autónoma de Madrid, 28029 Madrid, Spain; 3Instituto de Investigaciones Biomédicas “Alberto Sols”, CSIC-UAM, 28029 Madrid, Spain; 4Instituto de Investigación Hospital Universitario La Paz (IdiPAZ), 28029 Madrid, Spain; 5CIBER de Enfermedades Raras (CIBERER), Instituto de Salud Carlos III, 28029 Madrid, Spain

**Keywords:** meiosis, cilia, centrosome, mouse

## Abstract

Cilia are hair-like projections of the plasma membrane with an inner microtubule skeleton known as axoneme. Motile cilia and flagella beat to displace extracellular fluids, playing important roles in the airways and reproductive system. On the contrary, primary cilia function as cell-type-dependent sensory organelles, detecting chemical, mechanical, or optical signals from the extracellular environment. Cilia dysfunction is associated with genetic diseases called ciliopathies and with some types of cancer. Cilia have been recently identified in zebrafish gametogenesis as an important regulator of *bouquet* conformation and recombination. However, there is little information about the structure and functions of cilia in mammalian meiosis. Here we describe the presence of cilia in male mouse meiotic cells. These solitary cilia formed transiently in 20% of zygotene spermatocytes and reached considerable lengths (up to 15–23 µm). CEP164 and CETN3 localization studies indicated that these cilia emanate from the mother centriole prior to centrosome duplication. In addition, the study of telomeric TFR2 suggested that cilia are not directly related to the *bouquet* conformation during early male mouse meiosis. Instead, based on TEX14 labeling of intercellular bridges in spermatocyte cysts, we suggest that mouse meiotic cilia may have sensory roles affecting cyst function during prophase I.

## 1. Introduction

Centrioles are conserved microtubule-based intracellular structures [1,2]. Centrosomes are made up of two orthogonally barrel-shaped centrioles embedded in an electron-dense material called the pericentriolar matrix [3]. In mammals, each of the centrioles is composed of nine microtubule (MT) triplets around a central proteinaceous structure whose shape resembles a cartwheel [4]. In contrast, centrioles are composed of either MT triplets or MT doublets in flies [5] or a MT singlet in nematodes [6]. The function of centrosomes in most animal cells is crucial, since they behave as the MT organizing centers (MTOCs), controlling the configuration of the MT cytoskeleton of interphase cells, as well as regulating the formation of the bipolar spindle during cellular division [7,8,9,10]. For this reason, centrosome regulation is tightly coordinated with cell cycle progression [2]. In G1, the centrosome is composed of two parental centrioles, but it undergoes a duplication during the S phase, with each of the parental centrioles giving rise to a newly formed procentriole. Thus, by G2, cells contain two centrosomes, each consisting of an older and a younger centriole (known as the mother and daughter centrioles, respectively). Later, during mitosis, the two centrosomes occupy opposite poles of the dividing cell, where they shape the bipolar spindle [11].

The centrosome is also the cellular structure that shapes the axoneme, which is the template for the formation of cilia and flagella [3]. Cilia are MT-based plasma membrane projections that can function as motors and/or sensors. Motile cilia beat to displace extracellular fluids, as occurs in mammalian airways, reproductive tracts, and brain ventricles, where these cilia displace mucus, oocytes, and cerebrospinal fluid, respectively [12]. Flagella, despite their different name, are also, in essence, motile cilia, with the peculiarity that they are very long and move in a whip-like fashion, as occurs in sperm cells [13]. Motile cilia (and flagella) emanate from an axoneme consisting of nine peripheral MT doublets surrounding a central MT pair, an arrangement known as the 9 + 2 configuration. Along the peripheral MT doublets, the axonemal dynein molecules slide along adjacent MTs powering the motion of these organelles [3].

Unlike motile cilia and flagella, primary cilia function as cell-type-specific signaling organelles. They typically contain a 9 + 0 axoneme, lacking not only the central MT pair but also axonemal dynein, rendering these cilia immotile. Despite these differences, primary cilia are structurally similar to motile cilia. They also emanate from the mother centriole of the membrane-docked centrosome (known as a basal body in the ciliary context) and share many of the same markers, like acetylated tubulin (AcTub) and ARL13B [14,15,16]. Among the many important functions of primary cilia are: (i) embryonic Hedgehog signaling to pattern the nervous system and skeleton; (ii) monitoring of blood and urine flow; (iii) sensory perception of light, smell, and sound; and (iv) neuropeptide signaling in the brain to control behaviors such as feeding [17,18,19,20]. In accordance with their many important functions, motile and/or primary cilia malfunction is associated with genetic diseases called ciliopathies, which can be severe and affect multiple organs such as the eyes, kidneys, brain, lungs, or skeleton. In addition to ciliopathies, primary cilia are also involved in cancer, with constitutive activation of ciliary Hedgehog signaling being a common cause of tumorigenesis [21,22].

Cilium assembly (ciliogenesis) and disassembly are dynamically regulated with respect to cell cycle progression [23]. In G1/G0 somatic cells, the mother centriole presents subdistal and distal appendages, whereas the daughter centriole does not [24,25]. Subdistal appendages are involved in MT anchoring, however distal appendages contain proteins, like CEP164 (centrosomal protein of 164 kDa), that are essential for membrane docking of the mother centriole, thus forming the basal body and allowing subsequent ciliogenesis [24,26,27]. While primary cilia are typically stable in terminally differentiated cells, ciliated cycling cells must disassemble their cilia in G2, thereby releasing their centrioles from the ciliary base and allowing them to function as MTOCs at the spindle poles during cell division [28]. This is the case for several kinds of stem and progenitor cells, whose primary cilia regularly assemble and disassemble in each cell cycle [29].

While the organization and function of cilia have been studied in many somatic cell types, little is known about cilia´s role and presence in germ cells. Gametogenesis is the biological process that gives rise to gametes: oocytes in the case of female oogenesis and spermatozoa in the case of male spermatogenesis. Spermatogenesis comprises at least three functional processes: (i) proliferation of spermatogonia (stem germ cells), (ii) meiosis of spermatocytes, and iii) maturation of spermatids, which will be transformed into mature spermatozoa (spermiogenesis) [30]. Both meiosis and spermiogenesis require tight regulation of the cytoskeleton structures, particularly the centrosomes. Male meiosis is a specialized cell division that generates haploid gametes from diploid spermatogonia. It comprises two consecutive cell divisions following a single round of DNA replication [31,32]. To allow these two meiotic consecutive divisions, each one requires the assembly of a spindle, and therefore centrosomes must duplicate twice [33]. After meiosis, spermatids undergo dramatic morphological changes in order to transform into fully competent motile spermatozoa [34]. During this process, the sperm tail emerges from the mother centriole of the spermatid, which is thereby transformed into the basal body of the flagellum [35,36].

Ciliopathies are often associated with infertility [2,37]. Moreover, some studies have revealed that primary cilia dysfunction is associated with impaired male reproductive tract development. However, these effects were attributed to the malfunction of the primary cilium in somatic cells of the testicle, like Sertoli and peritubular myoid cells or cells of the epididymis [38]. So far, information about the presence of cilia in meiocytes is scarce. Two recent studies reported the presence of cilia in meiotic cells in zebrafish oogenesis [39] and spermatogenesis [40]. These reports indicated that the meiotic cilium might be an important regulator of chromosome polarization and recombination. In addition, these recent findings indicated that cilia might be present in male and female meiosis in mice [39], but a morphological or functional description is lacking.

This study presents the first thorough description of cilia in male mouse meiosis. We uncovered the precise stage at which the cilium is assembled, revealing that meiotic cilia are very transient structures. In addition, contrary to zebrafish, our results suggest that the role of the cilium in mammalian meiosis is not related to chromosome polarization.

## 2. Material and Methods

### 2.1. Materials

Testes from adult C57BL/6 (wild-type, WT) and genetically modified *Ankrd31*^−/−^ [41] male mice were used for this study. All animals’ procedures were approved by local and regional ethics committees (UAM Ethics Committee for Research and Animal Welfare) and performed according to European Union guidelines.

### 2.2. Squashing Procedure and Immunofluorescence Microscopy

Seminiferous tubules were fixed and processed following previously described protocols for the squashing technique [42].

Acetylated Tubulin was detected with a mouse monoclonal antibody (Sigma, T7461) at a 1:100 dilution. Tubulin was detected with a mouse monoclonal antibody (Abcam, ab7291) at a 1:100 dilution. Centrin 3 (CETN3) was detected with a mouse monoclonal antibody (Novus, H00001070-M01) at 1:100 dilution. ARL13B was detected with a mouse monoclonal antibody (Proteintech, 10002155) at a 1:30 dilution. SYCP3 was detected with either a mouse monoclonal antibody against mouse SYCP3 (Abcam, ab97672) or a rabbit polyclonal antibody recognizing mouse SYCP3 (Abcam, ab15093), both at a 1:50 dilution. SYCP1 was detected with a mouse monoclonal antibody against mouse SYCP1 (Abcam, ab15090) at a 1:50 dilution. TEX14 was detected with a rabbit polyclonal antibody (Proeintech, 18351-1-AP) at a 1:30 dilution. VASA was detected with a rabbit polyclonal against mouse DDX4/MVH (Abcam, ab13840) at a 1:100 dilution. TRF2 was detected with a rabbit polyclonal antibody (Novus, 56321) at a 1:50 dilution. Corresponding secondary antibodies were used against rabbit and mouse IgGs conjugated with either AMCA, Alexa 488, Alexa 555, or Alexa 647 (Molecular Probes); all were used at a 1:100 dilution.

Immunofluorescence images and stacks were collected on an Olympus BX61 microscope equipped with epifluorescence optics, a motorized Z-drive, and Olympus DP74 digital cameras controlled by Cellsens software (Olympus Life Science, Hachioji, Tokyo). Finally, images were processed with ImageJ (National Institute of Health, Bethesda, MD, USA; http://rsb.info.nih.gov/ij accessed on 11 September 2022) or/and Adobe Photoshop software.

### 2.3. Histology

For histological cryosections, dissected testes were included in OCT (Sakura Finetek Europe, AV Alphen aan den Rijn, The Netherlands) and immediately frozen. The OCT blocks were cut into 10 μm thick sections using a cryostat. The sample slides were allowed to dry at room temperature and hydrated in PBS three times, followed by fixation with 4% PFA for 10 minutes. Slides were washed in PBS twice and permeabilized in PBS/0.1% Tri-ton X-100 for 20 minutes. Slides were washed in PBS three times and blocked with PBS/2% BSA for 30 minutes before being used for immunofluorescence protocol.

### 2.4. Quantification Analysis

The presence of cilia was quantified in 100 zygotene spermatocytes of three biological replicates. The length of cilia was quantified in a minimum of 20 spermatocytes per stage. Statistical analysis was performed using a One-way ANOVA followed by Holm-Sidak’s post hoc test. Significant differences were considered when *p* < 0.05. The relation between cilia and *bouquet* configuration was quantified in a total of 50 spermatocytes. The number of cilia per cyst was quantified in a total of 25 cysts.

## 3. Results

### 3.1. Mouse Meiotic Cilia Are Present in Zygotene spermatocytes prior to Centrosome Duplication

Ciliary structures in mouse seminiferous tubules have been recently suggested to be very short projections, barely extending beyond the basal body [39]. However, the actual size, cycle of extension, and retraction, or even the precise stages at which cilia develop, remain unclear. We here present a detailed study of the localization and dynamics of the mouse’s meiotic cilium. For this purpose, we first studied the localization of the main component of its axoneme, acetylated tubulin (AcTub), during both meiotic divisions in squashed spermatocytes. We triple immunolabelled AcTub with a marker of centrioles, Centrin 3 (CETN3), and a marker of the progression of meiotic synapsis, the Synaptonemal Complex Protein 3 (SYCP3). CETN3 allowed us to determine whether centrioles have been duplicated, while SYCP3 allowed us to identify the meiotic stage, especially during prophase I.

At early prophase I, in leptotene, AcTub labeled the centrioles colocalizing with CETN3 (Figure 1A and A´). In the transition from leptotene to early zygotene, two alternative patterns could be seen. Some spermatocytes were not ciliated (Figure 1B and B´), while others showed an elongated hair-like structure labeled with AcTub, emanating from one of the CETN3 centriole signals (Figure 1C and C´). Quantitative analysis showed that around 20% of the spermatocytes at zygotene showed this hair-like structure (*n* = 100 spermatocytes at zygotene per individual, three biological replicates) (Figure 1D). This hair-like staining pattern of AcTub in spermatocytes at early zygotene is consistent with the solitary projection formed when primary cilia assemble in both somatic [14] and meiotic cells [39]. These zygotene cilia were present when only two centrioles were detected, which indicated that mouse meiotic cilia are assembled prior to centrosome duplication.

Centrioles duplicate during zygotene [33], as revealed by the presence of four different signals labeled with CETN3. Cilia were no longer observed in spermatocytes at late zygotene after centrioles had duplicated (Figure 1E and E´). At pachytene, AcTub was detected as a single signal per centrosome associated with the duplicated centrioles (Figure 1F and F´). At diplotene (Figure 1G and G´) and diakinesis (Figure 1H and H´-H´´), when centrosomes started migrating towards opposite poles, the signal of AcTub expanded to surround the two pairs of separated centrioles. During prometaphase I, AcTub still localized to the centrosomes (Figure 1I and I´–I´´). Once centrosomes had completely migrated to opposite poles and the meiotic bipolar spindle was established at metaphase I, AcTub labeled the two pairs of centrioles and the MTs of the bipolar spindle, while CETN3 only detected the centrioles (Figure 1J and J´). The same pattern persisted at anaphase I when homologous chromosomes segregated (Figure 1K and K´). At telophase I, CETN3 still labeled the centrioles, and AcTub labeled the centrioles and the midzone MTs (Figure 1L and L´). During interkinesis, the AcTub signal again accumulated at the centrosomes, as duplicated centrioles were clearly seen by the four CETN3 signals upon the second centrosome duplication (Figure 2A and A´). Cilia were not detected again prior to the second centrosome duplication (Figure 2E). At metaphase II (Figure 2B and B´) and anaphase II (Figure 2C and C´), AcTub was observed at the centrioles and the meiotic bipolar spindle II. At telophase II, AcTub labeled the centrioles and the midzone MTs (Figure 2D and D´).

Tubulin is a protein whose distribution has been extensively studied in mammalian meiosis. Therefore, it was surprising that the meiotic cilium was not observed before. In order to compare the distribution of AcTub with that of overall tubulin, we performed a colabeling with an antibody against αTubulin (αTub). A careful comparison of non-acetylated αTub and AcTub leads us to conclude that they show a different localization pattern in meiosis (Figure 3). At leptotene, αTub was mainly localized to cytoplasmatic MTs, while AcTub accumulated only at the centrosome (Figure 3A). In those zygotene spermatocytes in which the cilium was present, AcTub labeled this structure, while αTub was only located along cytoplasmatic MTs (Figure 3B). This indicates that αTub does not label meiotic cilia and may explain why these structures were not detected in previous studies. At metaphase I and metaphase II, both non-acetylated and acetylated tubulin labeled the spindle MTs, but only AcTub was detected at the centrosomes (Figure 3C,D).

To further characterize the organization and composition of the mouse meiotic cilium, we studied the localization of AcTub and ADP-ribosylation factor-like 13B (ARL13B) in histological cryosections of mouse testis. ARL proteins belong to the ARF family of GTPases, which are involved in ciliogenesis and ciliary intraflagellar transport (IFT) in somatic cells [43,44]. More specifically, ARL13B mediates primary cilia function and/or Hedgehog signaling [45]. In testis sections, spermatocytes located at the base of the semi-niferous tubule were the only ones that presented with cilia and an SYCP3 signal compatible with early prophase I (Figure 4I.(Aa–d)). However, AcTub also labeled flagella of spermatids facing the lumen of the seminiferous tubule (Figure 4I.(A)). ARL13B was present along the meiotic cilia visualized at prophase I spermatocytes located at the base of the tubule (Figure 4I.(Ba–d)). Triple immunolabeling of ARL13B, AcTub, and SYCP3 in squashed spermatocytes corroborated that ARL13B and AcTub colocalized in zygotene spermatocytes (Figure 4II.(A)).

These results confirm that the fiber-like structures observed at zygotene spermatocytes are bona fide cilia. Nevertheless, given that mouse flagella also incorporate AcTub [13], an additional corroboration was conducted in order to disprove that the cilia observed in zygotene could be mistaken with the flagella of nearby spermatozoa after the squashing procedure. For this, we studied the localization of AcTub in spermatocytes of *Ankrd31*^−/−^ [41], a sterile mouse model that lacks spermatozoa. Our results showed that *Ankrd31*^−/−^ presents zygotene cells with AcTub-labelled cilia (Appendix A), with a morphology similar to that of the wild type (WT) individuals (Appendix A). In contrast, only WT mice present early spermatids (Appendix A) and elongated mature spermatids (Appendix A) with AcTub-labelled flagella.

In conclusion, these results lead us to conclude that cilia are present in mouse spermatocytes at zygotene prior to centrosome duplication.

### 3.2. Meiotic Cilia Appear at the Onset of Synapsis

We next wanted to assess the temporal correlation between the formation of this structure and other typical meiotic processes, such as chromosome synapsis. For this, we triple immunolocalized AcTub with SYCP3 and SYCP1, a component of the transverse filaments of the synaptonemal complex [46]. At leptotene, when synapsis had not yet started and SYCP1 labeling was absent, AcTub was only detected at the centrioles (Figure 5I.(A)). The cilium was first observed in late leptotene spermatocytes when SYCP3-labeled axial elements were already formed, but SYCP1 protein was not yet loaded to chromosomes. The average length of the cilium at this stage was approximately 4,7 µm, presumably polymerizing (Figure 5I.(B),II.). Synapsis initiation marked the beginning of zygotene. At this stage, short filaments of SYCP1 were seen partially colocalizing with SYCP3. The cilium elongated conspicuously at this stage, reaching up to 22 µm, with an average length of 15µm at early-mid zygotene (Figure 5I.(C,D), and Figure 5II). When synapsis has partially progressed and longer filaments of SYCP1 were clearly seen (late zygotene), the meiotic cilium appeared to shrink, measuring an average of 9,3 µm long (Figure 5I.(E),II.). In the quantitative analysis of the cilia length (Figure 5II), One-way ANOVA analysis showed significant differences between the length of the cilia among the different stages (*p* < 0.0001). As previously indicated, the cilia were present in only a fraction (≈20%) of zygotene cells. Therefore, most spermatocytes at this stage did not present cilia. In those cases, AcTub only labeled centrioles (Figure 5F).

These results led us to conclude that mouse meiotic cilia start to polymerize at the transition from leptotene to zygotene, and they are fully formed by mid zygotene.

### 3.3. Mouse Meiotic Cilia Emanate from the Mother Centriole Prior to Centrosome Duplication

We then tested if mouse meiotic cilia emanate from the mother centriole. For this purpose, we first studied the dynamics of CEP164, a protein of the distal appendages required for cilia formation in somatic cells [24,47] that has also been identified at the mother meiotic centriole in mouse spermatocytes [48]. Our results corroborated that CEP164 labeled one of the two centrioles before centrosome duplication at early prophase I. At leptotene, CEP164 labeled a circular signal around one of the two CETN3-labeled centrioles of the not yet duplicated centrosome (Figure 6I.(A,A´)). Centrosome duplication occurs during zygotene [33,48], and four signals of CETN3 were observed, but CEP164 still labeled only one of the four centrioles at this stage (Figure 6I.(B,B´)). We then triple immunolabelled CETN3, CEP164, and AcTub. We found that the cilia emanated from the centriole that possessed CEP164 at early zygotene, i.e., from the mother centriole of the not yet duplicated centrosome (Figure 6II.(A)). The CETN3 signal did not overlap with AcTub cilium labeling, suggesting that when the cilium is fully formed, the centrioles are not acetylated (Figure 6II.(A´)). These results indicate that mouse meiotic cilia appear at late leptotene to early zygotene transition when centriole duplication has not yet occurred and that the cilium emanates from the mother centriole.

To corroborate the pattern of distribution of CEP164 and its implication during male mouse meiosis, we continued studying its localization during both meiotic divisions. We observed that CEP164 was sequentially loaded to the centrioles until the four centrioles assembled distal appendages at the end of the first meiotic division (Appendix A). Therefore, the four centrioles present CEP164 in spermatocytes at metaphase I and telophase I (Appendix A), according to previous results in mouse spermatocytes [48]. There was no incorporation of additional distal appendages in the daughter centrioles during the second meiotic division (Appendix A). CEP164 labeled only one of the two centrioles per centrosome in opposite poles of the bipolar meiotic spindle II (Appendix A). When spermiogenesis progressed, centrioles were disorganized in late spermatids, and no CETN3 signals were seen; however, the CEP164 signal was still present as previously reported [48,49], indicating the formation of a basal body for the growing flagellum (Appendix A).

Overall, our results prove that the meiotic cilium appears prior to the first centrosome duplication at zygotene. The cilium is not reassembled prior to nor during the second meiotic centrosome duplication at interkinesis.

### 3.4. Meiotic Bouquet Conformation and Cilia Formation Are Not Concurrent Events

Meiotic cilium in zebrafish oocytes favors the polarization of the chromosome telomeres, which are anchored to the nuclear envelope (NE), forming the so-called *bouquet* configuration [39]. To test whether the mouse zygotene cilium in spermatocytes is related to the *bouquet* conformation, we performed a double immunodetection of the telomeric double-stranded TTAGGG repeat binding proteins (TRF2) and AcTub. In spermatocytes at zygotene, we observed TRF2 as small and rounded signals randomly distributed, presumably over the NE, regardless of whether cilia labeled by AcTub were absent (Figure 6III.(A)) or present (Figure 6III.(B)). After triple immunolocalization of AcTub, SYCP3, and TRF2, only a semi-*bouquet* could be seen in a small fraction of spermatocytes at zygotene, where almost all telomeric ends were polarized to one region of the nucleus. This agrees with previous reports on mouse meiosis [50,51,52]. However, no cilia were observed in this situation (Figure 6III.(C)). Accordingly, no *bouquet* nor semi-*bouquet* configuration was presented in any ciliated spermatocyte at zygotene (Figure 6III.(D)). A graphical representation of the number of ciliated spermatocytes at zygotene (*n* = 50) that present the representative mouse semi-*bouquet* configuration is included (Figure 6IV). Therefore, there seems not to be a direct relationship between the presence of a cilium and the configuration of the telomeric *bouquet* in mouse spermatocytes.

### 3.5. Zygotene TEX14 Connected Cysts

Classical studies on the organization of the seminiferous tubules showed that spermatocytes form cysts of cells synchronously advancing through meiosis. These spermatocytes present cytoplasmic bridges that presumably provide biochemical and physiological coordination of the cells in a cyst [53,54]. As our results showed that only ≈20% of the spermatocytes at zygotene had a formed cilium, we then wanted to test if spermatocytes displaying the cilium could form specific cysts. For this, we detected TEX14, the product of testis-expressed gene 14 (Tex14), a protein that localizes to germ cell intercellular bridges in male spermatocytes [55]. Our results showed that TEX14 labeled ring-shaped intercellular bridges between spermatocytes at different stages. Cysts of early prophase I spermatocytes were seen at the base of the seminiferous tubules in testis cryosections (Figure 7I.(Aa)). In most of these spermatocyte cysts, the cilium could be detected in only one of the spermatocytes (Figure 7I.(Ab)).

The association of spermatocytes in cysts was also preserved in preparations of squashed seminiferous tubules. We double immunolabeled TEX14 and VASA, a germ-specific cytoplasmic marker [56], and corroborated a cytoplasmic continuity between the spermatocytes at the places where TEX14 bridges were present (Figure 7II.(A)). Our results suggested that cysts of connected spermatocytes presented mono-ciliation (Figure 7II.(B)). Furthermore, quantitative analysis showed that only one ciliated spermatocyte was observed per cyst (*n* = 25 cysts of spermatocytes at early zygotene) (Figure 7II.(C)). This demonstrates that cells bearing a cilium are not specifically connected in a cyst, but on the contrary, they preferentially form connections with other spermatocytes lacking the cilium.

## 4. Discussion

### 4.1. The Meiotic Cilium in Mouse Spermatogenesis

The primary cilium is an outgrowth of the plasma membrane located in many eukaryotic cell types. These immobile and solitary projections have diverse functions related to cell communication [57] and have traditionally been depicted as structures whose continued presence is incompatible with cell cycle progression [28]. However, although studies on the structure of primary cilia in various somatic tissues are extensive and diverse, their presence in vertebrate meiosis has only been reported in two recent works. The first one described the zygotene cilium in zebrafish oocytes, pointing to an evolutionary conservation of this structure in mammals, but suggested that ciliary structures in mouse spermatocytes were very short, barely extending beyond the basal body [39]. Our results demonstrate that the ciliary structures in mouse spermatocytes are not very short, on the contrary, cilia are strikingly large (up to 23 µm). This finding has been addressed by providing a detailed morphological analysis of adult mice in a reproductive stage, data that was lacking in that previous study [39]. The second study proposed that germ-cell-specific depletion of ciliary genes resulted in compromised double-strand break repair, reduced crossover formation, and increased germ-cell apoptosis in zebrafish spermatogenesis [40]. We here showed a detailed description of the mouse’s meiotic cilium, and we hypothesized that they are primary cilia. We found that around 20% of the spermatocytes at zygotene presented cilia as a long, solitary protrusion that disassembled once spermatocytes progressed into pachytene. This provides a scenario that challenges the assumption that primary cilia are antagonistic with cell division, as it has to be taken into account that cells at zygotene have already started meiotic cell division [58].

Apart from our work and the recent advances in zebrafish gametogenesis [39,40], another previous report also pointed to the presence of primary cilia during D. melanogaster spermatogenesis. Moreover, it suggested that these cilia were stable through both meiotic divisions in dividing spermatocytes with yet unknown functions [59]. Therefore, it appears that, although somatic primary cilia in vertebrates are sensory machinery during interphase, which was considered a “sleeping beauty” to date during cell division [60,61], cells that undergo meiosis are capable of polymerizing cilia once cell division has started. This may be related to the dynamics of the mitotic versus the meiotic centrosome. In somatic cells, centrosome duplication occurs, like DNA duplication, during the S phase, i.e., before mitosis [62]. In contrast, during gametogenesis, DNA duplication occurs in the premeiotic interphase, and in mice, centrosome duplication happens twice once meiosis begins, at zygotene, and again at interkinesis [33,48]. This implies that somatic centrosome duplication occurs after ciliogenesis and before entry into mitosis. In contrast, meiotic centrosome duplication occurs after ciliogenesis, with both processes taking place during meiosis.

### 4.2. Meiotic Centrosomes and Their Relation to Ciliogenesis and Flagelogenesis

Our results demonstrate that zygotene cilia in mouse spermatocytes contain acetylated tubulin -AcTub-, similar to the zygotene cilia in zebrafish spermatocytes and oocytes [39,40]. This was expected since this posttranslational modification of tubulin is a well-known marker in both motile and primary cilia in somatic cells [63,64,65]. As expected, we detected that tubulin was also acetylated in the flagellar axoneme of spermatids [13]. Given that the cilium emanates from the larger rounded signal of the two centrioles marked by CETN3, where CEP164 distal appendages are assembled, we concluded that zygotene cilia emanate from the mother centriole, as in somatic cells [66,67], pointing to conservation of the process of ciliogenesis between somatic and germ cells.

However, it is important to note that ciliogenesis from the mother centriole only occurs prior to the first centrosome duplication at zygotene, whereas another meiotic ciliogenesis from the new distal appendages does not occur prior to the second centrosome duplication at interkinesis. Our results reveal that CEP164 distribution was consistent with the function of distal appendages in cilia and flagella [68,69]. On the one hand, we show that CEP164 served to dock the basal body during zygotene ciliogenesis, sharing a similar role to its function in somatic cells [24,26,27]. On the other hand, spermatids presented CEP164 distal appendages to promote the polymerization of the flagellar axoneme, as previously reported [48]. Hence, the dynamics of CEP164 appear to be essential for the progression of spermatogenesis. This is consistent with the sterile phenotype of CEP164 deficient mice -FoxJ1-Cre;CEP164fl/fl- that showed sperm agglutination and aberrant ciliogenesis of motile multicilia in the somatic cells of the efferent ducts [49]. With the present work, we now suggest that CEP164 participates not only in flagelogenesis but also in mouse meiotic ciliogenesis. For further clarification, we present a scheme representative of the progression of male mouse meiosis in relation to centrosome dynamics, ciliogenesis, and flagelogenesis (Figure 8).

### 4.3. Tubulin Acetylation during Male Mouse Meiosis

We here showed that centrosomes presented AcTub partially colocalizing with CETN3 during the entire meiotic division, but nonacetylated α-Tub was never detected at centrosomes. CETN3 is a component of meiotic centrioles [33,48]; therefore, we can conclude that meiotic centrioles possess acetylated tubulin. This might promote their stability since experiments using biotinylated tubulin demonstrated that centriolar MTs do not exhibit significant exchange with the cytoplasmic protein pool [70]. This agrees with centriolar MTs being heavily acetylated and polyglutamylated, which are hallmarks of stable MTs [71]. In this sense, acetylation of MTs promotes the recruitment of PLK1 to the centrosomes in mitosis [72]. This kinase is located at meiotic centrosomes regulating centrosome maturation, separation, and migration [33,48]. Interestingly, our results also demonstrate that when meiotic cilia are formed, centrioles do not show AcTub. Future research should clarify the relationship between PLK1 and the acetylation of centriolar tubulin and the role this kinase might exert on meiotic ciliogenesis.

On the other hand, during mitotic metaphase, AcTub is enriched at interpolar and kinetochore MTs, but not at astral MTs, and AcTub becomes concentrated on the midbody during telophase and cytokinesis [73]. This acetylation is an evolutionarily conserved post-translational modification of α-Tub associated with the increased stability of various MT populations, including the mitotic spindle [65]. Moreover, tubulin acetylation does modulate the ability of mitotic MTs to bind to MAPs (Microtubule Associated Proteins) and motor proteins and may regulate MT stability and function [74,75]. Our results show that once the bipolar spindles are shaped at metaphase I and II, spindle MTs are acetylated. We also observed the labeling of AcTub in the midbody of telophase I and II. It has been described that the post-translational modifications of tubulin are related to the integrity of the spindle in oocytes [76] and the stability of the spermatozoa [13]. Specifically, an altered acetylation level of tubulin leads to defective spindle assembly and chromosome misalignment in mouse oocyte meiosis [77,78]. However, the function of AcTub in male mouse meiosis is not yet described. We suggest that the acetylation of tubulin in meiotic MTs might play similar roles, contributing towards the stability of the centrioles throughout the entire meiosis and towards the stability of the meiotic spindles I and II at metaphase I/II and the midbodies during telophase I/II.

### 4.4. Zygotene Cilia and Bouquet Formation Are Not Directly Related in Male Mouse Meiosis

From our results, it emerged that the cilium appeared in primary spermatocytes at the transition from leptotene to zygotene prior to centriole duplication. This indicates that the cell signaling processes in which the cilium may be involved might be related to the initiation of synapsis at zygotene [46,79], especially because cilia are no longer detected when synapsis is complete at pachytene. During zygotene, the *bouquet* polarization is a transient conformation of the chromosomal ends that favors the pairing process between homologous chromosomes [52]. However, this structure significantly differs between species. In zebrafish, the *bouquet* configuration is obvious and easily detected [80]; In C. *elegans*, the *bouquet* does not exist per se as the chromatin concentrates at one side of the nucleus adopting the characteristic half-moon shape [81]. In marsupial mammals, the bouquet stage is conspicuous [82], but in contrast, the *bouquet* chromosome polarization in mice is incomplete. Indeed, it behaves as a semi-*bouquet*, as it does not include all the chromosome ends, and it is extremely short-lived as it is only present in a time window limited to ∼0.5% of spermatocytes [50,51]. Recent reports in zebrafish oogenesis demonstrated a correlation between the zygotene cilia and the *bouquet* conformation, suggesting that the absence of cilia or their incorrect formation affects synapsis progression [39]. However, we could not find any direct correlation between the presence of cilia and *bouquet* conformation in mice. Our results indicate that the extremely short-lived *bouquet* in mice most likely precedes the formation of the cilia. Therefore, we conclude that there is no direct relation nor concurrency between the representative mouse semi-*bouquet* and the presence of meiotic cilia. Additionally, the zebrafish cilium has also been related to recombination regulation [40]. We cannot make any considerations in this regard, as the relationship between cilia and recombination in mice should require additional approaches.

### 4.5. Signaling Functions of the Cilia Could Be Spread to Cyst-Connected Spermatocytes

Primary cilia are found in many mouse cell types, and their morphology and size vary depending on their cellular functions. For example, primary cilia of cardiac fibroblasts are essential for heart development [83], and primary cilia of epithelial cells from the proximal renal tubule participate in the regulation of the glomerular filtration barrier [84].

In spermatocytes, we have observed the presence of a remarkably long cilium, approximately 15 µm, compared to the 6 µm described in zebrafish spermatocytes [39]. All aspects of cilium biology are regulated: their presence or absence on the cell surface, their morphology, their length, and ultimately their function [85]. Considering the structure we describe is a primary cilium, its length is similar to those of mouse renal epithelial cells, neurons, and olfactory cells [86,87]. In these somatic cells, primary cilia have a sensory function, capturing information from the extracellular environment and triggering an intracellular response. A greater surface area of the ciliary membrane could harbor more sensory transmembrane proteins and, therefore, more signaling receptors. That is why longer ciliary length has been related to greater sensitivity and response [88], and modifications of the ciliary length affect cilium integrity and, ultimately, its functions [86]. Given that the mouse meiotic cilium has a mean length equivalent to other long cilia in mouse somatic cells, we suggest that it may also perform important cell signaling functions during spermatogenesis. In parallel, we have also shown that mouse zygotene cilia present ARL13B, a GTPase of the ARL family known to mediate ciliary protein trafficking and regulate ciliary Hedgehog signaling [45]. Therefore, it is plausible that mouse zygotene cilia have an active Hedgehog pathway, which should be clarified in detail in future studies.

The histology of the testis might explain the absence of cilia in all spermatocytes. The organization of the epithelia of the seminiferous tubules is evolutionarily conserved in mammals and present cysts of cells undergoing spermatogenesis [89]. In mice, spermatogonia are linked by cytoplasmic bridges whose absence leads to infertility [54,55]. These bridges contain, among other proteins, the testis-expressed gene 14 product (TEX14) [55,90]. These bridges are partially released once spermatogonia evolve to primary spermatocytes upon entry to meiosis [91]. The proposed roles for the intercellular bridges include germ cell communication and synchronization [53] and sharing of gene products. An example of this function is the sharing of proteins encoded by Akap genes of the X chromosome between the four spermatids resulting from meiosis so that they can all form the fibrous sheath of the sperm [92]. The fact that only 20% of the spermatocytes at zygotene present cilia leads us to suggest that in a cyst of spermatocytes at zygotene with cytoplasmic continuity, not all the spermatocytes need to be ciliated to progress through spermatogenesis. Therefore, this leads us to hypothesize that spermatocytes within the same cyst could share the signaling events provided by a single ciliated spermatocyte. For further clarification, a scheme representing the seminiferous epithelia is presented (Figure 9).

In conclusion, we argue that the need to maintain intercellular connections in spermatocyte cysts for achieving fertility [53], together with the unique anchoring junctions in the testis —Sertoli-Sertoli, Sertoli-spermatogonia, Sertoli-spermatocyte and Sertoli-spermatid cell unions [93,94]— might require complex signaling that helps the progres-sion of cells from base to lumen of the seminiferous tubule during spermatogenesis and subsequent spermiogenesis (Figure 9). One might think that the long zygotene cilia could be implicated in the signaling pathway between spermatocytes in coordination with Sertoli cells, at least during the first stages of prophase I. Given that ciliopathies present various anomalies in the assembly or regulation of the ciliary and flagellar axoneme [95], more studies will be necessary to analyze the relationship between meiotic cilia and sterility.

The present work has described the zygotene cilium in mouse spermatocytes suggesting cyst monociliation in the seminiferous epithelium, opening the line of research for future studies to unravel the importance of meiotic ciliogenesis.

## Figures and Tables

**Figure 1 cells-12-00142-f001:**
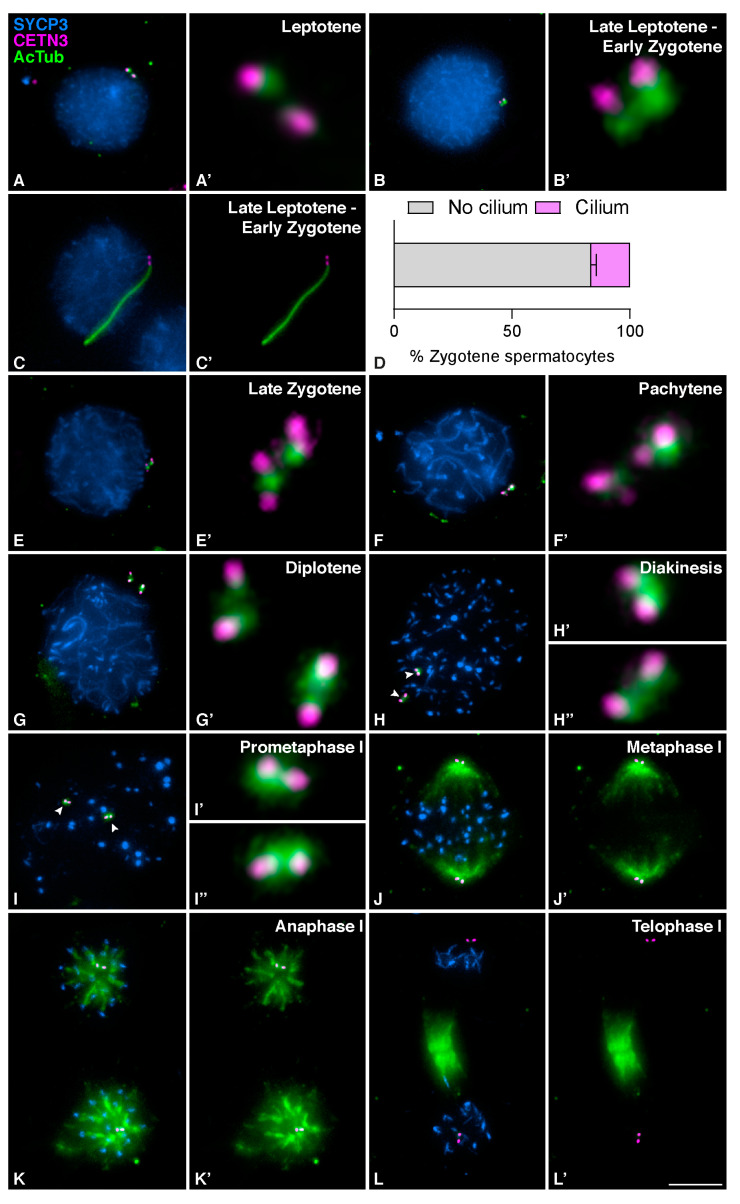
**Mouse spermatocytes form transient cilia during meiotic prophase I.** Detection of acetylated tubulin in mouse spermatocytes during the first meiotic division. Triple immunolabelling of synaptonemal complex protein 3 (SYCP3) (blue), Centrin 3 (CETN3) (magenta), and acetylated Tubulin (AcTub) (green) on squashed WT mouse spermatocytes at (**A**) Leptotene, (**B**) Leptotene to zygotene transition, Late leptotene-Early Zygotene, without cilium, (**C**) Late Leptotene-Early Zygotene, with cilium, (**D**) Graphical representation of the quantification of ciliated spermatocytes at Zygotene (*n* = 100, three biological replicates), (**E**) Late Zygotene, (**F**) Pachytene (**G**), Diplotene, (**H**) Diakinesis, (**I**) Prometaphase I, (**J**) Metaphase I, (**K**) Anaphase I, and (**L**) Telophase I. For images (**A**′–**I**′) the 300× magnification for centrosomes is shown ((**H**′,**H″**) and (**I**′,**I″**) white arrowheads). Scale bar in (**L′**) represents 5 µm.

**Figure 2 cells-12-00142-f002:**
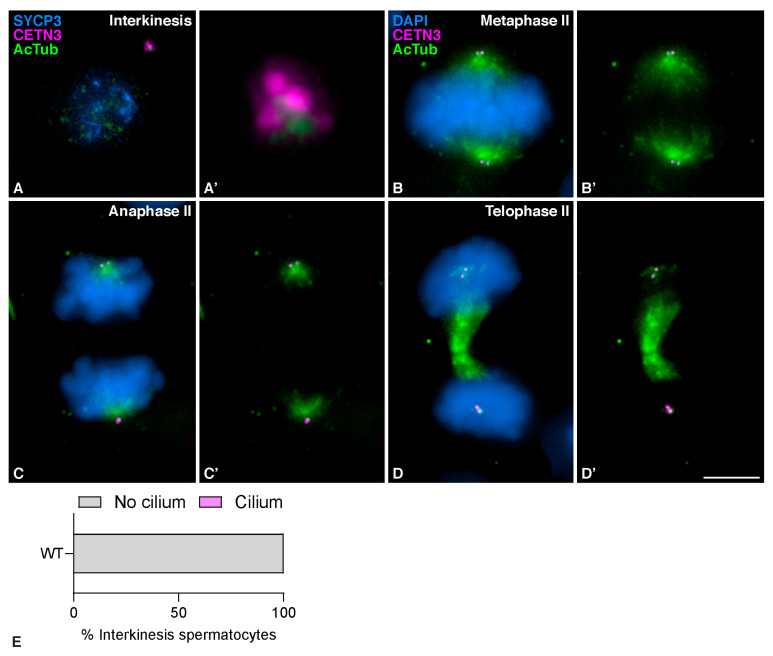
**Mouse secondary spermatocytes do not present cilia.** Detection of acetylated Tubulin in mouse spermatocytes during the second meiotic division. Triple immunolabelling of synaptonemal complex protein 3 (SYCP3) (blue), Centrin 3 (CETN3) (magenta) and acetylated Tubulin (AcTub) (green) on squashed WT mouse spermatocytes at (**A**) Interkinesis. And double immunolabelling of Centrin 3 (CETN3) (magenta) and acetylated Tubulin (AcTub) (green), with chromatin stained with DAPI (blue) at (**B**) Metaphase II, (**C**) Anaphase II (**D**) and Telophase II. For images (**A′**) the 300× magnification of the centrosomes is shown. (**E**) Graphical representation of the quantification of ciliated spermatocytes at interkinesis (*n* = 100, three biological replicates). Scale bar in (**D′**) represents 5 µm.

**Figure 3 cells-12-00142-f003:**
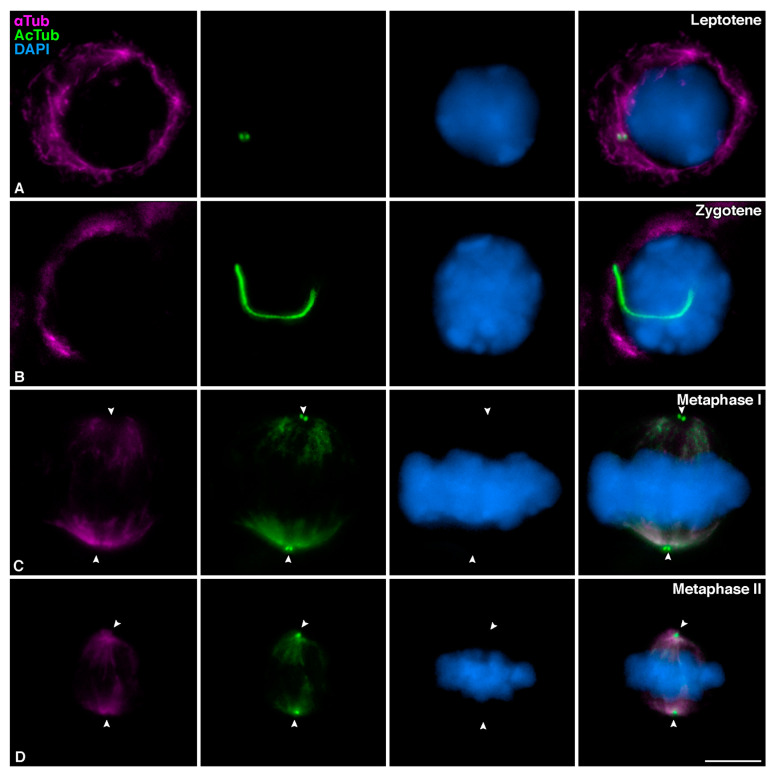
**Study of the comparison between acetylated tubulin and non-acetylated tubulin in mouse spermatocytes.** Double immunolabelling of α tubulin (αTub) (magenta) and acetylated tubulin (AcTub) (green), with chromatin stained with DAPI (blue) on squashed WT mouse spermatocytes at (**A**) Leptotene, (B) Zygotene with primary cilium, (**C**) Metaphase I, and (**D**) Metaphase II. White arrowheads in (**C**,**D**) indicate centrosomes localization. Scale bar in (**D**) represents 5 µm.

**Figure 4 cells-12-00142-f004:**
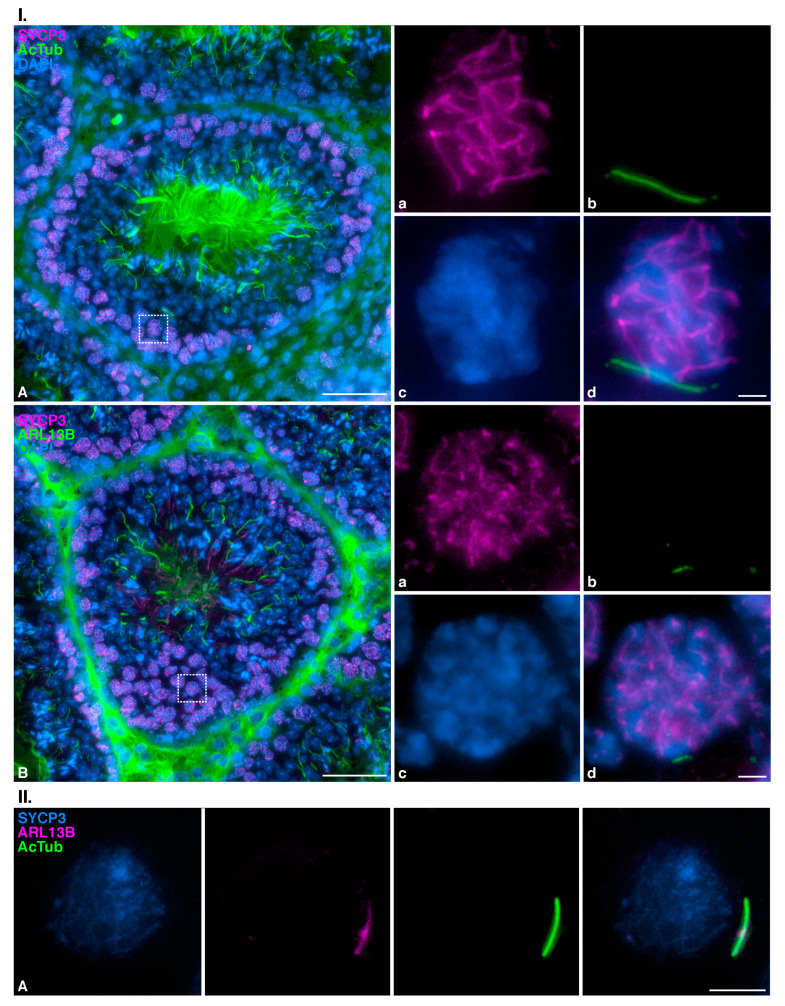
**Zygotene cilia of mouse testis spermatocytes contain ARL13B.** (**I**)**: Distribution of acetylated Tubulin and ARL13B in mouse testis cryosection.** Double immunolabelling of SYCP3 (magenta) and acetylated Tubulin (AcTub) (green), with chromatin stained with DAPI (blue) on cryosections of mouse testis. (**A**) Complete section of a seminiferous tubule, (**a**–**d**) magnified selected spermatocyte at zygotene (dotted square) showing a fully formed primary cilium, and double immunolabelling of SYCP3 (magenta) and ARL13B (green), with chromatin stained with DAPI (blue) on cryosections of mouse testis. (**B**) Complete section of a seminiferous tubule, (**a**–**d**) magnified selected spermatocyte at zygotene (dotted square) showing an incipient polymerizing primary cilium. Scale bar in A and B represents 50 µm, and scale bar in d represents 5 µm. (**II**)**: Distribution of ARL13B in mouse squashed spermatocytes at zygotene.** Triple immunolabelling of SYCP3 (blue), ARL13B (magenta), and acetylated tubulin (AcTub) (green) on squashed WT mouse spermatocytes at zygotene (**A**). Scale bar represents 5 µm.

**Figure 5 cells-12-00142-f005:**
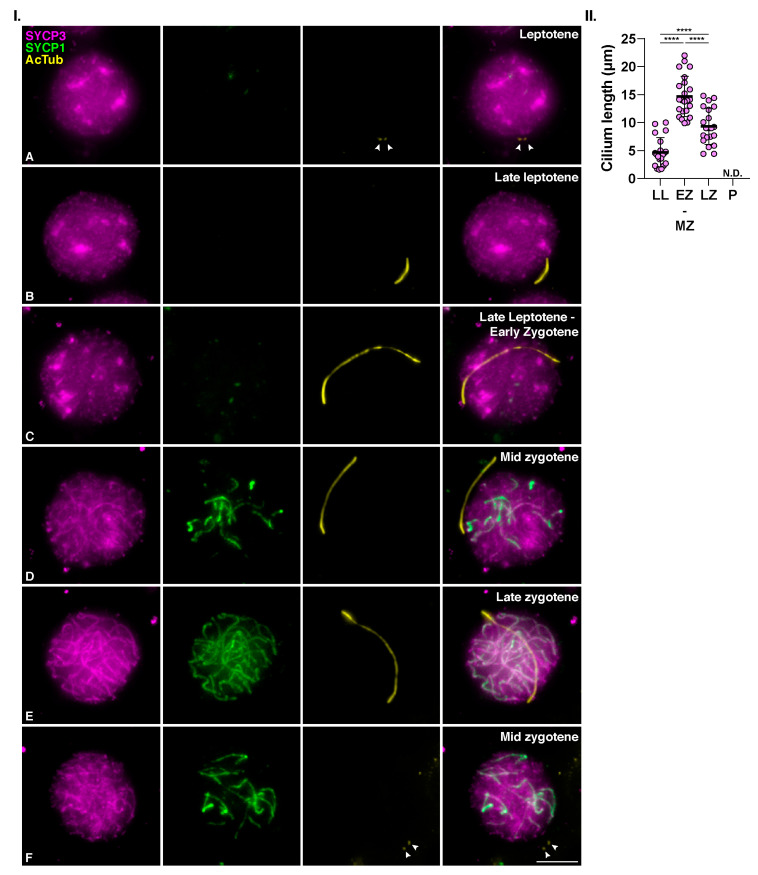
**Mouse meiotic cilia appear at the onset of synapsis.** (**I**)**. Cilia are fully formed at mid zygotene.** Triple immunolabelling of SYCP3 (magenta), SYCP1 (green), and acetylated tubulin (AcTub) (yellow) on squashed WT mouse spermatocytes at (**A**) Leptotene, (**B**) Late Leptotene with polymerizing primary cilium, (**C**) Leptotene to zygotene transition, Late leptotene—Early Zygotene with primary cilium, (**D**) Mid Zygotene with primary cilium, (**E**) Late Zygotene with primary cilium, and (**F**) Mid zygotene without cilium. White arrowheads in A and F indicate centrosomes localization. Scale bar in F represents 5 µm. (**II**)**. Cilia length quantification.** Graph represents the length of the cilia in spermatocytes at Late Leptotene, Early-Mid Zygotene, Late Zygotene, and Pachytene. Data represent mean ± SD, **** *p* < 0.0001, One-way ANOVA, a minimum of 20 spermatocytes quantified per zygotene stage, while *n* = 100 pachytene spermatocytes were analyzed, and none of them were ciliated.

**Figure 6 cells-12-00142-f006:**
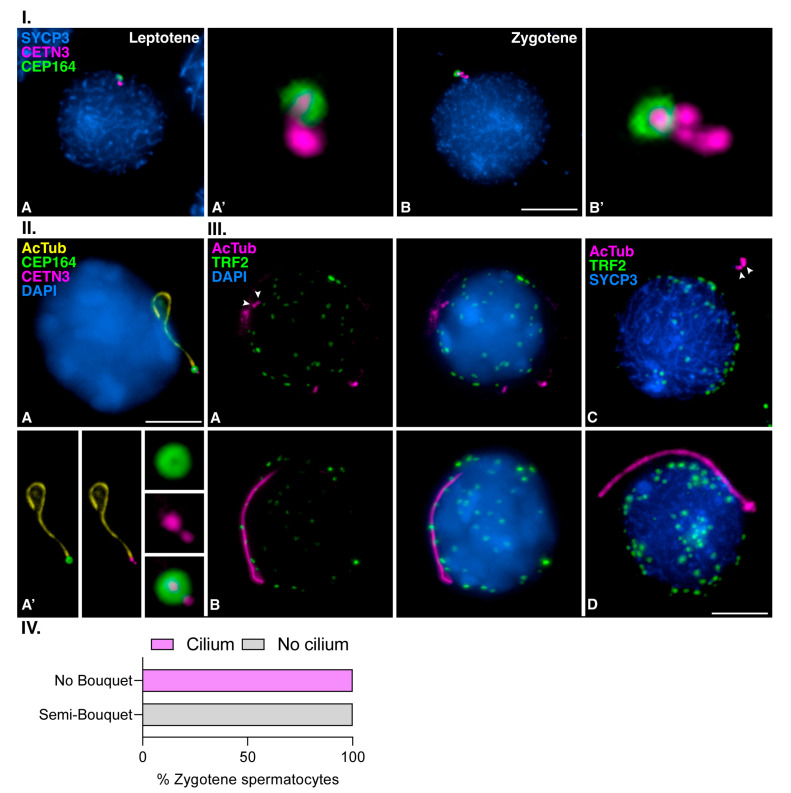
**Mouse meiotic cilia emanate from the mother centriole before centrosome duplication.** (**I**)**. Distribution of CEP164 in early prophase I spermatocytes;** Triple immunolabelling of SYCP3 (blue), Centrin 3 (CETN3) (magenta), and CEP164 (green) in mouse spermatocytes at (**A**) Leptotene, and (**B**) Zygotene. Images (**A′**,**B′**) show the 300× magnification of the centrosomes. Scale bar in B represents 5 µm. (**II**)**. Meiotic zygotene cilia emanate from the unduplicated mother centriole;** Triple immunolabelling of acetylated Tubulin (AcTub) (yellow), CEP164 (green), and Centrin 3 (CETN3) (magenta), with chromatin stained with DAPI (blue) on squashed WT mouse spermatocytes at (**A**) Zygotene presenting a primary cilium. Images in (**A′**) show the primary cilia and the 300× magnification of the centrosomes. Scale bar in A represents 5 µm. (**III**)**. The presence of the primary cilia is not directly related to the bouquet conformation of chromosome ends;** Double immunolabelling of acetylated Tubulin (AcTub) (magenta) and TRF2 (green), with chromatin stained with DAPI (blue) on squashed WT mouse spermatocytes at (**A**) Zygotene without cilium, and (**B**) ciliated Zygotene. Triple immunolabelling of acetylated tubulin (AcTub) (magenta) and TRF2 (green) and SYCP3 (blue) on squashed WT mouse spermatocytes at (**C**) Zygotene without cilium and (**D**) Zygotene with cilium. White arrowheads in (**C**,**D**) indicate centrosomes localization. Scale bar in (**D**) represents 5 µm. (**IV**)**. There is no direct relation nor concurrency between the presence of meiotic cilia and the representative mouse *bouquet* configuration;** Graphical representation of the quantification of ciliated spermatocytes at Zygotene (*n* = 50, three biological replicates) that present a semi-*bouquet* conformation.

**Figure 7 cells-12-00142-f007:**
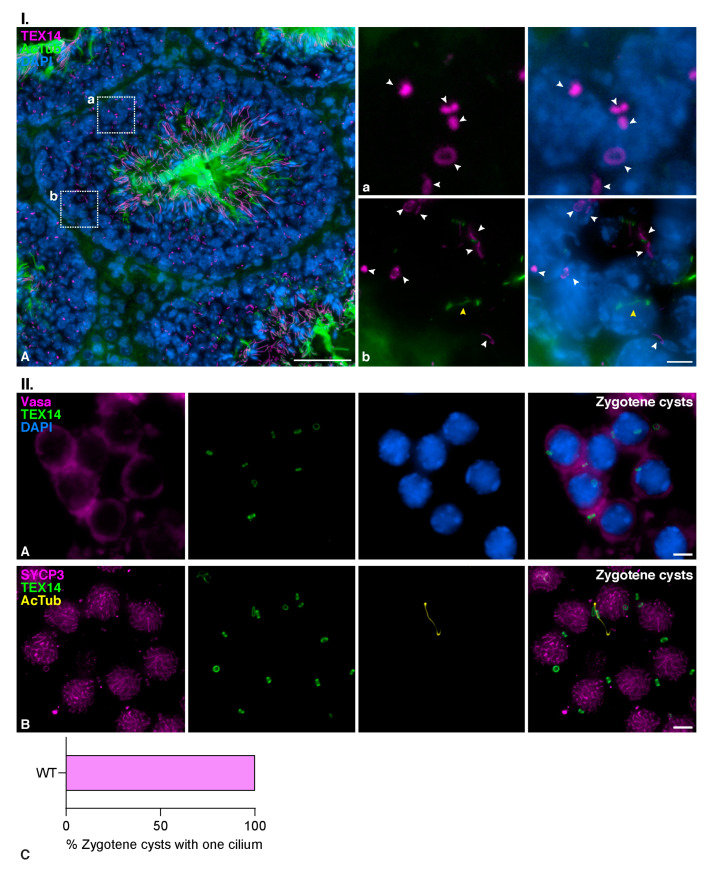
**Spermatocyte cysts share cilia among several spermatocytes at zygotene.** (**I**)**. Distribution of TEX14 cellular bridges in mouse testis cryosections;** Double immunolabelling of TEX14 (magenta) and acetylated tubulin (AcTub) (green), with chromatin stained with DAPI (blue) on cryosections of mouse testis. (**A**) Complete section of a seminiferous tubule. Scale bar in A represents 50 µm. (**a**) Magnified selected (dotted square) cyst of primary spermatocytes interconnected by TEXT14 bridges (white arrowheads), and (**b**) magnified selected (dotted square) cyst of primary spermatocytes interconnected by TEXT14 bridges (white arrowheads), with one zygotene showing a primary cilium (yellow arrow). White arrowheads in a and b indicate TEX14 bridges. Scale bar in b represents 5 µm. (**II**)**. Distribution of TEX14 cellular bridges in mouse spermatocytes at zygotene;** (**A**) Double immunolabelling of VASA (magenta) and TEX14 (green), with chromatin stained with DAPI (blue) on squashed WT mouse spermatocytes. A cyst of spermatocytes at zygotene interconnected by TEXT14 bridges is shown. Scale bar represents 5 µm. (**B**) Triple immunolabelling of SYCP3 (magenta), TEX14 (green), and acetylated tubulin (AcTub) (yellow). A cyst of spermatocytes at zygotene interconnected by TEXT14 bridges is shown, with one zygotene showing a primary cilium. Graphical representation of the quantification of the number of ciliated spermatocytes per cyst at zygotene (*n* = 25 cysts). Scale bar represents 5 µm.

**Figure 8 cells-12-00142-f008:**
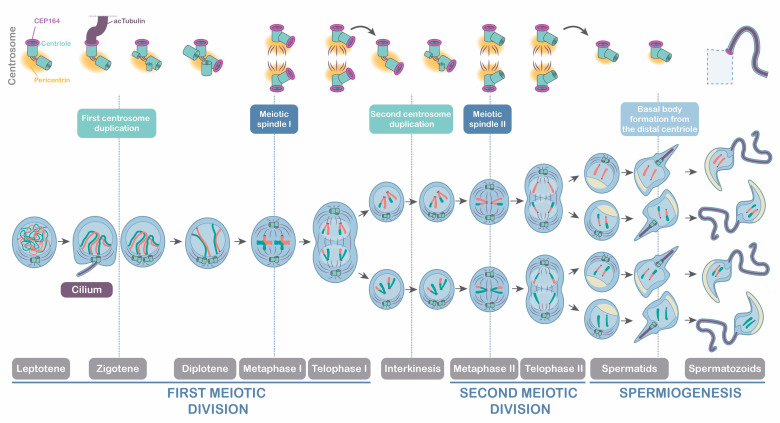
**Schematic representation of the meiotic stages in relation to centrosome dynamics.** Meiosis progression showing the centrosomal events (centrosome duplication I and II and formation of meiotic spindle I and II). Ciliogenesis and flagelogenesis events are represented: centrosome duplication occurs at zygotene transition and interkinesis; the formation of primary cilia occurs at zygotene transition; and flagellum is formed in spermatids. The distribution of centrioles, pericentriolar matrix (Pericentrin), and distal appendages (CEP164) are represented through both meiotic divisions. Primary cilia at zygotene and flagella at spermatids are shown with acetylated tubulin (AcTub). A schematic representation of spermiogenesis shows the formation of the acrosome and the flagella of the spermatozoids (dotted box).

**Figure 9 cells-12-00142-f009:**
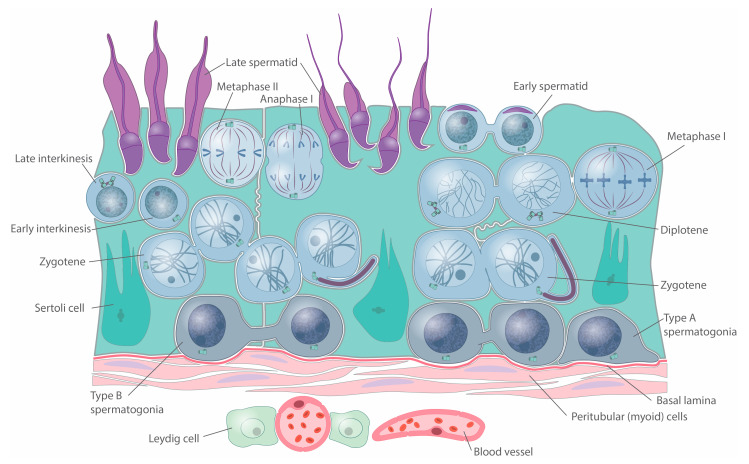
**Schematic representation of the seminiferous epithelium with cyst monociliation.** Scheme represents spermatogonia (dark grey), spermatocytes (grey), and spermatids (purple) embedded on the cytoplasm of Sertoli cells (turquoise). Centrosomes are indicated for spermatogonia and spermatocytes. Cilia are represented in one of the zygotenes of a cyst of primary spermatocytes. Peritubular and interstitial compartments are represented, indicating the position of myoid cells, Leydig cells, and blood vessels.

## Data Availability

The raw data for immunofluorescence images and statistical data supporting the conclusion of this article will be made available by the authors upon request.

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
