# Peer review of "The Male Mouse Meiotic Cilium Emanates from the Mother Centriole at Zygotene Prior to Centrosome Duplication"

_cells, 2022, doi:10.3390/cells12010142_

Round 1

Reviewer 1 Report (Previous Reviewer 1)

The authors have addressed most of my questions. However, I still have some minor comments that should be addressed before the manuscript is accepted for publication in Cells:

1.     There is no affiliation of the senior author on the cover page of the ms.

2.     Lines 33-35: As I pointed out in the comment 4 of my first review, in some eukaryotes, the centriolar MTs are organized in doublets or singlets, instead of the triplets.

3.     Lines 39-40: Strictly speaking, there are two centrosomes—not one centrosome—in G1 cells. During the transition from M to G1 phase, both centrioles of the two nascent cells convert to centrosomes.

4.     Lines 46-48: This sentence is not accurate. The “9+2 configuration” is characteristic of the motile cilia, in which centrosomes are usually inactivated.

5.    Line 84: “about its role” – “about cilia’s role”.

6.    Line 85: “give rise” – “gives rise”.

7.    Line 159: “spermatocytes during the second meiotic division” – As far as I understand, Figure 1 refers to the first (and not the second) meiotic division.

8.    Line 164: Add a comma after “Pachytene”.

9.    Line 174: Add “, and” after “Anaphase II”.

10. Line 210: “and 4 II.Aa-d” – The panels (a-d) are not labeled in Fig. 4II.

11. Line 218: “amplified selected spermatocyte” – Change to “Magnified selected spermatocyte (dotted square)”. Also, throughout the ms, it seems more appropriate to use the word “magnified”, instead of “amplified”, in analogous contexts (E.g., lines 374-375 etc). 

12. Line 246, 249, and 251. It would be easier for the reader if, after mentioning Fig. 5I, a reference to Fig 5II is also given in parentheses here, in the main text. 

13. Lines 251-253: “A quantitative analysis of the cilia length is presented for spermatocytes at late leptotene, early-mid zygotene, and late zygotene (Figure 5 II) (n=minimum of 20 cells per stage).” – I suggest to move this sentence to Figure 5’s legend.

14. Line 262: “Table 3. (magenta)” - does not seem to belong here. Change to: SYCP1 (green), SYPC3 (magenta), and acetylated Tubulin (AcTub) (yellow)…”.

15. Line 266 and throughout the ms: “White arrows” – these are arrowheads, not arrows. Please correct this throughout the ms (E.g. also lines 306, 376 etc.).

16. Lines 308-311: The graph in Fig. 6IV is misleading. As far as I understand, there should be four columns on this graph: the percentage of ciliated vs non-ciliated cells among cells without a bouquet vs those with a semi-bouquet.

17. Lines 335-336: “cilia labelled by AcTub is present (Figure 6 III.A) or absent (Figure 6 III.B).” – It should be vice versa: “cilia labelled by AcTub is ABSENT (Figure 6 III.A) or PRESENT (Figure 6 III.B).”

Author Response

We would like to enhance our gratitude to Referee #1 for her exhaustive review of our manuscript, which has polished it up to high standards.

Please notice that all referee´s comments are copied in blue. Author´s responses are in black.

The changes are highlighted in green in the revised version (word document with tracked changes)

The authors have addressed most of my questions. However, I still have some minor comments that should be addressed before the manuscript is accepted for publication in Cells: 

  1. There is no affiliation of the senior author on the cover page of the ms. Corrected
  2. Lines 33-35: As I pointed out in the comment 4 of my first review, in some eukaryotes, the centriolar MTs are organized in doublets or singlets, instead of the triplets. The reviewer is right, we previously mentioned eukaryotes but specific information about flies and nematodes is now provided in the text. We thank the reviewer for insisting in this important specification and apologise for not addressing it before.
  3. Lines 39-40: Strictly speaking, there are two centrosomes—not one centrosome—in G1 cells. During the transition from M to G1 phase, both centrioles of the two nascent cells convert to centrosomes. The reviewer is right, but we believe that the sentence “In G1, the centrosome is composed of two parental centrioles” is correct and more detail could lead to confusion.
  4. Lines 46-48: This sentence is not accurate. The “9+2 configuration” is characteristic of the motile cilia, in which centrosomes are usually inactivated. The reviewer is right, and now this sentence has been moved and only referees to motile cilia (lines 56-65).
  5. Line 84: “about its role” – “about cilia’s role”. Corrected
  6. Line 85: “give rise” – “gives rise”. Corrected
  7. Line 159: “spermatocytes during the second meiotic division” – As far as I understand, Figure 1 refers to the first (and not the second) meiotic division. Corrected
  8. Line 164: Add a comma after “Pachytene”. Corrected
  9. Line 174: Add “, and” after “Anaphase II”. Corrected
  10. Line 210: “and 4 II.Aa-d” – The panels (a-d) are not labeled in Fig. 4II. Corrected, text now reads “Figure 4 II” (line 186). As this image shown different immunobeling of the same cells, we don’t think is necessary to include a-d identifications to each image.
  11. Line 218: “amplified selected spermatocyte” – Change to “Magnified selected spermatocyte (dotted square)”. Also, throughout the ms, it seems more appropriate to use the word “magnified”, instead of “amplified”, in analogous contexts (E.g., lines 374-375 etc). This has been corrected in the entire manuscript, we now use the word “magnified” instead of “amplified”. We thank the reviewer for this accurate correction.
  12. Line 246, 249, and 251. It would be easier for the reader if, after mentioning Fig. 5I, a reference to Fig 5II is also given in parentheses here, in the main text. Included.
  13. Lines 251-253: “A quantitative analysis of the cilia length is presented for spermatocytes at late leptotene, early-mid zygotene, and late zygotene (Figure 5 II) (n=minimum of 20 cells per stage).” – I suggest to move this sentence to Figure 5’s legend. We have simplified the text and transfer the information about the number of cells analysed to the legend of Figure 5.
  14. Line 262: “Table 3. (magenta)” - does not seem to belong here. Change to: SYCP1 (green), SYPC3 (magenta), and acetylated Tubulin (AcTub) (yellow)…”. We can’t find what the reviewer refers to in this suggestion, but we confirm that we have reviewed the legend of Figure 5, which is now correct.
  15. Line 266 and throughout the ms: “White arrows” – these are arrowheads, not arrows. Please correct this throughout the ms (E.g. also lines 306, 376 etc.). This has been corrected in the entire manuscript, we now use the word “arrowheads” instead of “arrows”.
  16. Lines 308-311: The graph in Fig. 6IV is misleading. As far as I understand, there should be four columns on this graph: the percentage of ciliated vs non-ciliated cells among cells without a bouquet vs those with a semi-bouquet. We thank the reviewer for this suggestion. However, we don’t believe that four columns are needed. Instead, we have corrected the graph and now it says “semi-bouquet” instead of “bouquet”. As mentioned in the first response to the Reviewer 1, and the second response to the reviewer 2, a full bouquet is hardly or never achieved in mouse, therefore the graph should refer to the semi-bouquet. The results section in the main text has been adapted accordingly (lines 269-275, word with tracked changes).
  17. Lines 335-336: “cilia labelled by AcTub is present (Figure 6 III.A) or absent (Figure 6 III.B).” – It should be vice versa: “cilia labelled by AcTub is ABSENT (Figure 6 III.A) or PRESENT (Figure 6 III.B).” We are grateful to the reviewer for detecting this mistake, which is now corrected.

We are deeply thankful to reviewer 1 for his/her exhaustive reading of our manuscript and for providing excellent feedback, which has improved our work.

Reviewer 2 Report (Previous Reviewer 2)

I am now satisfied with the authors latest revision. I still have an issue with how impactful this study will be. but they have addressed my major concerns. I recommend publication. Thanks for the opportunity to review in Cells.

Author Response

We are happy that the revised version of the manuscript fulfils reviewer’s suggestions, and thank him/her for the helpful comments. We hope that this work could be beneficial to the scientific community.

Reviewer 3 Report (New Reviewer)

In this manuscript, authors presented well defined and scientifically sound research regarding the presence of cilia in male mouse meiotic cells, also uncovered the stage at which the cilium is assembled, suggesting that meiotic cilia are very transient structures. 

The Title of the manuscript is concise and relevant. The aim and scope of the study explained well. The abstract is easy to understand and well written. Introduction is quite comprehensive and highlighted work importance as well as its significance towards future prospective. Materials and methods are quite descriptive. Authors provided well explained interpretation of results and discussion. Overall, this research article is nicely written with sound performed descriptive experiments.

Before proceeding further, I expect the authors to thoroughly proofread the document and fix all grammatical and typographical errors (some examples include L35, L101, L575 etc).

Minor suggestions:

1)      L47, L398-401: font consistency should be maintained.

2)      L101/103: Avoid mentioning the same reference again and again, if reference points from the same reference lined up back to back. One time denotation is enough.

3)      Please provide ‘Institutional Review Board Statement’, as this research includes animal studies (refer author’s instructions).

4)      Author contribution and funding should be separated (refer author’s instructions).

Author Response

Please notice that all referee´s comments are copied in blue. Author´s responses are in black.

The changes are highlighted in blue in the revised version (word document with tracked changes)

We are thankful for reviewer 3 positive comments on our work. We are happy that he/she finds our manuscript satisfactory. We provide the explanation for the corrections required below.

Minor suggestions:

1)      L47, L398-401: font consistency should be maintained. Corrected

2)      L101/103: Avoid mentioning the same reference again and again, if reference points from the same reference lined up back to back. One time denotation is enough. Corrected, we have deleted the duplication of references, especially the one mentions for reference 37.

3)      Please provide ‘Institutional Review Board Statement’, as this research includes animal studies (refer author’s instructions). The UAM Ethics committee was already mentioned in the previous version, but we now follow the author´s instructions for Cells journal and have included the information in the required format, including the protocol code.

4)      Author contribution and funding should be separated (refer author’s instructions). This has been corrected, and funding is now a separate section.

This manuscript is a resubmission of an earlier submission. The following is a list of the peer review reports and author responses from that submission.

Round 1

Reviewer 1 Report

This manuscript by López-Jiménez and colleagues provides evidence that a subset of mouse male meiotic cells, spermatocytes, form a primary cilium at zygotene stage. The authors further suggest that this structure is dispensable for the meiotic chromosome dynamics but has a sensory/signaling role to enable proper spermatogenesis.

Overall, the study is interesting and well performed and presented. 

I have some critical comments that should be addressed before the manuscript can be further considered for publication. 

Major points:

1.     Somewhat contradictory to the report by Mytlis et al. (Ref. 36), the authors suggest that the role of the cilium in spermatocytes is “not directly related to the bouquet conformation”.  However, this suggestion may seem unfounded because the actual meiotic chromosomal bouquet is not shown in the manuscript. I assume that there are technical difficulties in detecting the bouquet in mice because, as the authors state in lines 435-436, the bouquet conformation “is extremely short-lived in this species”. I wonder if the authors were able to detect the chromosomal bouquet in mouse spermatocytes at all? If yes, can the authors carry out a statistical comparison of the proportion of ciliated spermatocytes between the populations of cells with a chromosomal bouquet vs those lacking the bouquet? If the authors cannot detect the bouquet at all, do the ciliated cells always have a randomly distributed TRF2 signal?

2.     In lines 307-308, the authors state that in most of the spermatocyte cysts, “the cilium could be detected in only one of the spermatocytes”. I wonder if this is always the case? Can the authors present quantitative data showing that there is consistently only one ciliated spermatocyte per cyst? If so, can the authors speculate if this fact in itself may give us possible clues to the function of the primary cilium in spermatogenesis?

Minor points:

1.     In several figures (Fig. 1, Fig. 2, Fig. 3, Fig. 4-II), the coloring of subcellular structures on the IF images does not match that in the figure legends. Please make sure that these errors are corrected everywhere. 

2.     Lines 211-213 and throughout the manuscript: The authors provide incorrect reference to the supplementary figures. E.g. on line 211, ‘(Figure A1 A,B)’, while it should be ‘(Figure S1 A,B)’. Please make sure that all these errors are corrected. 

3.     Lines 31-32: “Centrioles are conserved microtubule-based intracellular structures that form the core of the centrosome and act as templates for the formation of cilia and flagella [1, 2].”– Centrioles are a part of the centrosome. In many terminally differentiated ciliated or flagellated cells the centrosomes are inactivated. The authors should clarify the distinction between the terms ‘centriole’ and ‘basal body’. The same applies to lines 57-60.

4.     Lines 34-35: “Each of the centrioles is composed of nine microtubule (MT) triplets…” – In some organisms, the centriolar MTs are organized in doublets or singlets, instead of the triplets.

5.     Lines 36-37: “The function of centrosomes in animal cells is absolutely essential…”. This statement is not accurate: there are animal cells and even whole animal organisms that lack centrosomes.

6.     Lines 87-88: “Meiosis is a specialized cell division that generates haploid gametes from diploid spermatogonia.” – Meiosis also occurs during female gametogenesis. 

7.     Lines 95-97: “Moreover, some studies have revealed that dysfunction of primary cilia is associated with impaired male reproductive tract development.” – Add a reference(s).

8.     Lines 118-120: “Some spermatocytes showed elongated hair-like structure labelled with AcTub, emanating from one of the CETN3 centriole signals (Figure 1B and B´), while some others presented AcTub only at the centrioles (Figure 1C and C´).” - Figures 1B and B’ and 1C and C’ actually show different stages of meiosis (leptotene to zygotene transition vs late zygotene, respectively). 

9.     Line 131: Decipher the abbreviation for SYCP3.

10.  Line 146: “At telophase I, CETN3 still labels the centrosomes” – ‘centrioles’, not ‘centrosomes’?

11.  Line 147: “AcTub labels the centrioles and the midzone MTs (Figure 1J and J´)” - In Fig 1J and J’, AcTub does not seem to localize to centrosomes.

12.  Line 179: Add a comma after ‘cilium’. 

13.  Lines 190-191: “in zygotene spermatocytes (Figure 4 II.A-D).” - In Figure 4 II, only panel (A) is labeled.

14.           Line 197: “polymerized primary cilium” – Here and throughout the manuscript, the authors apply the term ‘polymerization’ to the cilium. I do not think this is accurate because, in addition to the polymerized microtubules, the cilia also contain many other components. I, therefore, believe that it is more appropriate to use other terms, such as ‘cilium formation’ or ‘cilium growth’, instead of ‘cilium polymerization’.

15.           Lines 211-212: “with a similar morphology than wild type (WT) individuals” – “with a morphology similar to that of the wild type (WT) individuals”?

16.           Lines 221-222: “At leptotene, when synapsis has not yet started and SYCP1 labelling is absent, AcTub is only detected at the centrioles (Figure 5A).” – In Fig. 5A, CYCP1 is not entirely absent; there are two dots of CYCP1.

17.           Line 252: delete ‘we’ after ‘then’.

18.           Line 274: “CEP64” – It should be “CEP164”.

19.           Line 278: “according to previous results spermatocytes [44].” A sentence-construction error?

20.           Lines 279-281: “CEP164 labeled only one of the two centrioles per centrosome in opposite poles of the bipolar meiotic spindle II.” – Does this sentence refer to Fig. S3? Again, please make sure that the supplementary figures are correctly referred to throughout the manuscript.

21.           Line 282: “is still present” – “was still present”.

22.           Line 284: “(Figure A2 A-E).” – Figure S3, I assume?

23.           Lines 298-300: “These spermatocytes present cytoplasmic bridges that presumably provide a biochemical and physiological coordination of the cells in a cyst.” – Provide a reference(s).

24.           Line 301: “have a polymerized cilium” – “formed a cilium”?

25.           Line 376: “to their function” – “to its function”?

26.           Line 393: “is represented” – “are represented” or “are shown”.

27.           Lines 404-405: “This kinase is located at meiotic centrosomes regulating their migration [30, 44].” – PLK1 also regulates centrosome separation and maturation.

28.           Lines 475-476: “Scheme represents spermatogonias (dark blue), spermatocytes (blue)” – To me, in Fig. 9, the spermatogonia (not spermatogonias) and spermatocytes look grey rather than blue.

Author Response

Please notice that all referee´s comments are copied in blue. Author´s responses are in black. The numbers referred in the details below correspond to the pages and lines of the revised version with tracked changes.

Response to the Referee #1

This manuscript by López-Jiménez and colleagues provides evidence that a subset of mouse male meiotic cells, spermatocytes, form a primary cilium at zygotene stage. The authors further suggest that this structure is dispensable for the meiotic chromosome dynamics but has a sensory/signaling role to enable proper spermatogenesis.

Overall, the study is interesting and well performed and presented.

I have some critical comments that should be addressed before the manuscript can be further considered for publication.

Author response: We are hugely grateful to referee #1 for sharing his/her expertise and offering positive comments for the major points. We are also grateful for reading our manuscript with so much detail and sharing with us all this minor comments concerns. His/her suggestions have helped to improve our work. We respond to all concerns below.

Major points:

  1. Somewhat contradictory to the report by Mytlis et al. (Ref. 36), the authors suggest that the role of the cilium in spermatocytes is “not directly related to the bouquet conformation”.  However, this suggestion may seem unfounded because the actual meiotic chromosomal bouquet is not shown in the manuscript. I assume that there are technical difficulties in detecting the bouquet in mice because, as the authors state in lines 435-436, the bouquet conformation “is extremely short-lived in this species”. I wonder if the authors were able to detect the chromosomal bouquet in mouse spermatocytes at all? If yes, can the authors carry out a statistical comparison of the proportion of ciliated spermatocytes between the populations of cells with a chromosomal bouquet vs those lacking the bouquet? If the authors cannot detect the bouquet at all, do the ciliated cells always have a randomly distributed TRF2 signal?

We truly appreciate this suggestion and thank the reviewer for recommending it.

Our techniques allow to visualize every stage of male mouse meiosis and spermiogenesis. However, the bouquet conformation of chromosome ends in mouse meiosis is an extremely transient stage and it does not include all chromosomal ends (1). In spermatocytes at zygotene, it is easy to find cells with a partial confluence of chromosomal ends, but the full bouquet configuration is extremely rare. Previous reports pointed that telomere clustering occurs in a time window limited to the onset of zygotene, which only leaves a few such cells detectable (∼0.5% of spermatocytes) for a very short period of time (2). This is clearly different to other species like zebrafish, where the bouquet configuration is more obvious and easily detected (3,4); C. elegans, where the bouquet does not exist per se but the chromatin concentrates at one side of the nucleus and adopts a prominent and characteristic half-moon shape (5); or even in other mammals like marsupials, where we have described a conspicuous bouquet (6). Therefore, different organisms present distinct chromosome configurations during prophase I, presenting differences particularly relevant during oogenesis (7). However, many questions still remain unclear about the specification of the bouquet configuration in spermatogenesis.

Our squash technique allows us to visualize the cells in a 3D configuration, and it is common to detect most of the telomere ends in one area of the nucleus but also some of them remaining outside this semi bouquet configuration at zygotene. Therefore, it is possible to observe a partial bouquet, where most, but not all, of the chromosome ends are located in a particular area of the nuclear envelope. To prove this, we have included a new image in Figure 6, where a partial bouquet conformation (labelled by TRF2) is observed in a spermatocyte at zygotene (Fig. 6 II.C,D). In this image, and in all the spermatocytes at an equivalent semi bouquet arrangement, there is no presence of the meiotic cilia. If there was a direct correlation between the presence of bouquet and the meiotic cilia, we would observe the cilia in these partial bouquet conformations, especially because our results demonstrate that cilia start at the onset of synapsis (Figure 5), where no full bouquet is observed.

To complete our results, we have developed a quantification assay in n=20 ciliated spermatocytes at zygotene and none of them presented a bouquet configuration of chromosome ends. This is included in revised lines 267-271 of the results section. The legend of Figure 6 III also includes this quantification data.

1.   Enguita-Marruedo, A., et al., Transition from a meiotic to a somatic-like DNA damage response during the pachytene stage in mouse meiosis. PLoS Genet, 2019. 15(1): p. e1007439.

2.   Liebe, B., et al., Mutations that affect meiosis in male mice influence the dynamics of the mid-preleptotene and bouquet stages. Exp Cell Res, 2006. 312(19): p. 3768-81.

3.   Siegfried, K.R., It starts at the ends: The zebrafish meiotic bouquet is where it all begins. PLoS Genet, 2019. 15(1): p. e1007854.

4.   Imai, Y., et al., Meiotic Chromosome Dynamics in Zebrafish. Front Cell Dev Biol, 2021. 9: p. 757445.

5.   Link, J. and V. Jantsch, Meiotic chromosomes in motion: a perspective from Mus musculus and Caenorhabditis elegans. Chromosoma, 2019. 128(3): p. 317-330.

6.   Page, J. et al., The pairing of X and Y chromosomes during meiotic prophase in the marsupial species Thylamys elegans is maintained by a dense plate developed from their axial elements. J. Cell Sci. 2003, 116, p551-560.

7.   Rubin, T., N. Macaisne, and J.R. Huynh, Mixing and Matching Chromosomes during Female Meiosis. Cells, 2020. 9(3).

  1. In lines 307-308, the authors state that in most of the spermatocyte cysts, “the cilium could be detected in only one of the spermatocytes”. I wonder if this is always the case? Can the authors present quantitative data showing that there is consistently only one ciliated spermatocyte per cyst? If so, can the authors speculate if this fact in itself may give us possible clues to the function of the primary cilium in spermatogenesis?

We understand the concern of the referee and provide here an explanation.

Yes, we only observe one ciliated spermatocyte per cyst. We now present quantitative data for 25 cysts of intact groups of cells were observed after the squashing procedure, and where TEX14 bridges where clearly seen between spermatocytes.

We cannot exclude the possibility that in cysts composed of a high number of cells, two ciliated spermatocytes could exist, yet we have never detected them.

In this regard, previous reports described the presence of spermatocyte cysts and suggested that these bridges are partially released once spermatogonia evolve to primary spermatocytes upon entry to meiosis (Geenbaum et al 2006; Li et al 2013) – cite in the original submission-. Therefore, the exact number of cells that compose a cyst is still undetermined, and one could suggest that some cysts contain more spermatocytes than others. However, we have never seen any cyst with more than one ciliated spermatocyte. This quantification data is included in the results section in revised lines 290-293. The legend of Figure 7 II also includes this quantification data.

We believe that the special cell adhesions between Sertoli-Sertoli, Sertoli-spermatogonia, Sertoli-spermatocyte and Sertoli-spermatid cell unions (including gap junctions) (Su et al 2013; Qian et al 2014) might favor an active cell communication between ciliated spermatocytes and the ones that lack the meiotic cilium. Therefore, we believe that the important conclusion to be drawn from our results is that, in a cyst of spermatocytes at zygotene, where cytoplasmic continuity is demonstrated by the presence of TEX14 bridges, not all the spermatocytes are ciliated. We hypothesize that this could provide spermatocyte synchronization and the sharing of gene products between spermatocytes of the same cyst. This discussion is now included in revised lines 423-439.

Minor points:

  1. In several figures (Fig. 1, Fig. 2, Fig. 3, Fig. 4-II), the coloring of subcellular structures on the IF images does not match that in the figure legends. Please make sure that these errors are corrected everywhere. We apologize for this mistake that of course has now been corrected in all legend of figures.
  2. Lines 211-213 and throughout the manuscript: The authors provide incorrect reference to the supplementary figures. E.g. on line 211, ‘(Figure A1 A,B)’, while it should be ‘(Figure S1 A,B)’. Please make sure that all these errors are corrected. 

We are sorry for this inconvenience. We consulted previous articles published in Cells and interpreted that supplementary Figures were meant to be named as “Figure AX”. This is now corrected to “Figure SX” in the entire manuscript.

  1. Lines 31-32: “Centrioles are conserved microtubule-based intracellular structures that form the core of the centrosome and act as templates for the formation of cilia and flagella [1, 2].”– Centrioles are a part of the centrosome. In many terminally differentiated ciliated or flagellated cells the centrosomes are inactivated. The authors should clarify the distinction between the terms ‘centriole’ and ‘basal body’. The same applies to lines 57-60.

We are very grateful to the reviewer for this suggestion. This information is now included in the introduction (revised lines 57-59).

  1. Lines 34-35: “Each of the centrioles is composed of nine microtubule (MT) triplets…” – In some organisms, the centriolar MTs are organized in doublets or singlets, instead of the triplets.

We thank the review for this annotation. It is now corrected in revised line 45, and a new reference is included (Carvalho-Santos Z, et al. Stepwise evolution of the centriole-assembly pathway. J Cell Sci. 2010 May 1;123(Pt 9):1414-26. doi: 10.1242/jcs.064931)

  1. Lines 36-37: “The function of centrosomes in animal cells is absolutely essential…”. This statement is not accurate: there are animal cells and even whole animal organisms that lack centrosomes. The referee is right, this statement is now corrected in revised line 47.
  2. Lines 87-88: “Meiosis is a specialized cell division that generates haploid gametes from diploid spermatogonia.” – Meiosis also occurs during female gametogenesis. 

To avoid misunderstandings, we have included a general introductory sentence at the beginning of this paragraph. The rest of the paragraph applies only to spermatogenesis (lines 95-97)

  1. Lines 95-97: “Moreover, some studies have revealed that dysfunction of primary cilia is associated with impaired male reproductive tract development.” – Add a reference(s).

The reference was included in the next sentence (Girardet et al, 2019), but is now correctly placed in the revised line 111.

  1. Lines 118-120: “Some spermatocytes showed elongated hair-like structure labelled with AcTub, emanating from one of the CETN3 centriole signals (Figure 1B and B´), while some others presented AcTub only at the centrioles (Figure 1C and C´).” - Figures 1B and B’ and 1C and C’ actually show different stages of meiosis (leptotene to zygotene transition vs late zygotene, respectively). 

The referee is very right in this suggestion. Figure 1 is now revised and presents, in addition to previous data, a spermatocyte at zygotene with unduplicated centrioles and no cilia. In the revised version of this figure, panels B and C now correspond to equivalent stages.

  1. Line 131: Decipher the abbreviation for SYCP3. This is now included in revised line 135. A short justification of the use of SYCP3 and CETN3 is also included.
  2. Line 146: “At telophase I, CETN3 still labels the centrosomes” – ‘centrioles’, not ‘centrosomes’? Yes, the referee is correct. This sentenced has been corrected in revised line 161.
  3. Line 147: “AcTub labels the centrioles and the midzone MTs (Figure 1J and J´)” - In Fig 1J and J’, AcTub does not seem to localize to centrosomes. Centrioles are labelled with AcTub in telophase I, but it seems that the intensity of the signal is lower. For further clarification, we provide for the referee the separated channels below. To avoid overintensity of the midbody in the merged image, centrioles are barely detected (please see image in the annexxed pdf).
  1. Line 179: Add a comma after ‘cilium’. It is now corrected in revised line 180.
  2. Lines 190-191: “in zygotene spermatocytes (Figure 4 II.A-D).” - In Figure 4 II, only panel (A) is labeled. This has been corrected now to (Figure 4 II.A and 4 II. Aa-d) in revised lines 191.
  3. Line 197: “polymerized primary cilium” – Here and throughout the manuscript, the authors apply the term ‘polymerization’ to the cilium. I do not think this is accurate because, in addition to the polymerized microtubules, the cilia also contain many other components. I, therefore, believe that it is more appropriate to use other terms, such as ‘cilium formation’ or ‘cilium growth’, instead of ‘cilium polymerization’. We thank the review for this suggestion, which we believe accurate. It is now corrected in the entire manuscript (please see an example in revised line 225 or 241).
  4. Lines 211-212: “with a similar morphology than wild type (WT) individuals” – “with a morphology similar to that of the wild type (WT) individuals”? This has been corrected in revised line 198-200.
  5. Lines 221-222: “At leptotene, when synapsis has not yet started and SYCP1 labelling is absent, AcTub is only detected at the centrioles (Figure 5A).” – In Fig. 5A, CYCP1 is not entirely absent; there are two dots of CYCP1. We believe that those faint signals correspond to background. The size of the spermatocyte nucleus, and the pattern of localization of SYCP3 allows a very clear recognition of the shown spermatocyte stage as leptotene.
  6. Line 252: delete ‘we’ after ‘then’. We believe that the referee refers to the typo in previous line 235. This has been corrected in revised line 237.
  7. Line 274: “CEP64” – It should be “CEP164”. We thank the reviewer. This has been corrected in revised line 244.
  8. Line 278: “according to previous results spermatocytes [44].” A sentence-construction error? We thank the reviewer for detecting this. This has been corrected in revised line 249.
  9. Lines 279-281: “CEP164 labeled only one of the two centrioles per centrosome in opposite poles of the bipolar meiotic spindle II.” – Does this sentence refer to Fig. S3? Again, please make sure that the supplementary figures are correctly referred to throughout the manuscript.

We apologize for not indicating the figures correctly. This is now corrected in revised line 247-249.

  1. Line 282: “is still present” – “was still present”. Corrected in revised line 253.
  2. Line 284: “(Figure A2 A-E).” – Figure S3, I assume? Corrected in revised line 255.
  3. Lines 298-300: “These spermatocytes present cytoplasmic bridges that presumably provide a biochemical and physiological coordination of the cells in a cyst.” – Provide a reference(s). Classic studies by Dym et al (1971), and more recent Greembaum et al (2011) publications are included in revised line 278.
  4. Line 301: “have a polymerized cilium” – “formed a cilium”? This has been corrected in revised line 279. The term polymerized cilium has been changed to formed cilium in the entire text.
  5. Line 376: “to their function” – “to its function”? Corrected in revised line 340.
  6. Line 393: “is represented” – “are represented” or “are shown”. Corrected in revised legends of Figures 8 and 9.
  7. Lines 404-405: “This kinase is located at meiotic centrosomes regulating their migration [30, 44].” – PLK1 also regulates centrosome separation and maturation. Corrected in revised line 360.
  8. Lines 475-476: “Scheme represents spermatogonias (dark blue), spermatocytes (blue)” – To me, in Fig. 9, the spermatogonia (not spermatogonias) and spermatocytes look grey rather than blue. We apologize for this misunderstanding and agree with the reviewer that “grey” is a better color to describe the diagram. This is now corrected in revised legend of Figure 9.

We hope that the new data of the revised version, together with the discussion, are of interest to the reviewer. We are very grateful for the exhaustive revision of our manuscript.

Reviewer 2 Report

The manuscript by Lopez-Jimenez et al. entitled “The male mouse meiotic cilium emanates from the mother centriole at zygotene prior to centrosome duplication” attempts to further characterize the recent discovery of primary cilium in mouse spermatocytes during zygotene in meiosis I. The authors are able to reproduce the most basic results found by Mytlis et al. Science (2022) regarding primary cilia in mouse spermatocytes and present some novel observations about this process; the timing of cilia growth in zygotene, the lack of correlation between cilia formation and bouquet formation, and the presence of one cilia per spermatocyte cyst. Unfortunately, these are the only novel contributions in this paper and a complete absence of quantitative data makes it impossible to accept these characterizations with any confidence.

The majority of the conclusions reached by the authors throughout the paper have been reported elsewhere.

Figure 1: authors report ~20% of spermatocytes have cilia in early zygotene, no quantification is given for this claim. Spermatocytes are assigned very specific phases of meiosis (e.g. early zygotene, late zygotene) with little explanation to the classification and with no apparent markers to confirm the classification. Primary cilia in mouse spermatocytes at zygotene has been previously reported (Mytlis et al. Science (2022), Li, Xinhua et al. Cells (2021)). Authors also report the presence of acetylated tubulin on the spindle and midbody, this is already known.

Figure 2: authors report no primary cilia in meiosis II and the presence of acetylated tubulin on the spindle and midbody, again these are already known.

Figure 3: authors report that not all tubulin in a cell is acetylated and that αTubulin is not a reliable marker for primary cilia, these are both already known.

Figure 4: authors report that SYCP3 positive spermatocytes have primary cilia. This is reported in Mytlis et al. Science (2022) and does not add any new information about the timing of cilia formation to the data the authors have already presented, though it is a more reliable piece of data as it uses a marker to show spermatocyte’s stage of meiosis and uses an additional cilia marker.

Figure 5: authors report that the primary meiotic cilium in mouse spermatocytes starts at 5µm in length at the beginning of zygotene, grows up to 15µm and then shrinks to 10-13µm at mid-late zygotene. No quantification is provided to support these claims. Authors should include a graph showing the measured lengths of primary cilia at early, mid, and late zygotene and show that there is a significant difference in lengths of cilia in cells at different stages in zygotene.

Figure 6: authors report that the mother centriole, as marked by Cep164, forms the primary cilium in mouse spermatocytes. This has already been defined as the mechanism for primary ciliogenesis (Graser et al. JCB (2007), Ishikawa & Marshall Nat Rev Mol Cell Bio (2011)). Authors also report that cilia are not present during bouquet formation during meiosis. This is contrary to the model presented by Mytlis et al. and is novel. However, the authors provide no quantification on how the presence of primary cilia in spermatocytes correlates with the presence of a bouquet, nor do they show an example of a spermatocyte with a bouquet that lacks a cilium as a control, so it is impossible to draw this conclusion from the data.

Figure 7: authors report that spermatocytes are connected through cytoplasmic bridges (already known). Authors also report that there is only one primary cilia per spermatocyte cyst and speculate that the function of the primary cilia is coordinating meiosis within a cyst. While this is an intriguing model the authors provide no quantification for numbers of primary cilia per cyst and provide no evidence for this mechanic action beyond an observed correlation, so it is impossible to draw this conclusion from the data.

Novelty: While the question addressed here is well defined and relatively original the results presented here do not contribute anything new to answer the question.

Scope: The paper does fit within the scope of the journal.

Significance: Without any quantification it is unclear whether the results are significant. The conclusions drawn from the data are overstatements of what the data actually show.

Quality: The article is wordy in an attempt to expand a small body of results into a full article. The authors present well established results that they reproduce as novel findings which is misleading.

Scientific Soundness: Experiments are presented without any controls or quantification and as a result the data presented are neither robust nor scientifically sound. The methods are sparse.

Overall Merit: There is no overall benefit to publishing this work.

English Level: The English is understandable.

Ultimately, I recommend that this manuscript be rejected as it makes no original contribution to the field and even major revisions would not increase its potential impact.

Author Response

Please notice that all referee´s comments are copied in blue. Author´s responses are in black. The numbers referred in the details below correspond to the pages and lines of the revised version with tracked changes.

Response to the Referee #2

The manuscript by Lopez-Jimenez et al. entitled “The male mouse meiotic cilium emanates from the mother centriole at zygotene prior to centrosome duplication” attempts to further characterize the recent discovery of primary cilium in mouse spermatocytes during zygotene in meiosis I. The authors are able to reproduce the most basic results found by Mytlis et al. Science (2022) regarding primary cilia in mouse spermatocytes and present some novel observations about this process; the timing of cilia growth in zygotene, the lack of correlation between cilia formation and bouquet formation, and the presence of one cilia per spermatocyte cyst. Unfortunately, these are the only novel contributions in this paper and a complete absence of quantitative data makes it impossible to accept these characterizations with any confidence.

The majority of the conclusions reached by the authors throughout the paper have been reported elsewhere.

We thank the reviewer for reading and providing feedback for our manuscript. As he/she points out, previous recent data have reported the presence of cilia in mouse meiosis in Mytlis et al (Science, 2022). In this excellent publication, authors focused on the female and male meiosis of the model zebrafish and present an outstanding description of the primary cilia in both spermatogenesis and oogenesis of that species. In Mytlis et al (2022), Figure 6 presents mouse oogenesis, and Supplementary Figure 19 presents mouse spermatogenesis. Page 10 of Mytlis et al (2022) reads “We also identified ciliary structures in spermatocytes in postnatal day 12 and 16 mouse testes using multiple ciliary markers including Arl13b and GluTub co-labeled with Sycp3 (fig. S19). Adcy3 further identified these ciliary structures and colocalized to the centrosome basal body as labeled with gTub (fig. S19A). These ciliary structures in mouse spermatocytes were very short and included mostly a shortly extended basal body (arrowheads in fig. S19, right panels)” (Mytlis et al, 2022).

This being said, we respectfully disagree with some of the reviewer statements posed in relation to our work.

First, our results do not reproduce the work of previous authors. In fact, we refute one of the main statements of the report presented by Mytlis and coworkers: the ciliary structures in mouse spermatocytes are not very short, as these authors stated, but, on the contrary, cilia are strikingly large (up to 20µm). This finding was only possible by providing detailed morphological analyses that were lacking in previous studies.

Second, the reviewer himself/herself points out to some of our highlights (ciliogenic timing, non-correlation with bouquet, and cyst monociliation) as original contributions. We agree that our work provides these new findings, plus we consider that there are still other relevant contributions. Therefore, we sincerely consider that our study is pertinent and necessary. Some of these additional findings are:

  • We present a detailed description of the meiotic cilia in male mouse meiosis. Moreover, some of our results question that the model posed for zebrafish could be applied to mammals. This is expected as cellular mechanisms of different organisms might behave differently.
  • We present data extracted from adult mice in a reproductive stage, rather than prepuberal 12-16 dpp mice (Mytlis et al, 2022).
  • We present precise ciliogenic timing in both meiotic divisions: We identify the presence of the meiotic cilium in zygotene, i.e., early prophase I of the first meiotic division. In male mouse meiosis there are two centrosome duplication rounds, the first one in zygotene, and the second one in interkinesis. Our results prove that the meiotic cilium appears prior to the first centrosome duplication at zygotene, whereas the cilium is not reassembled prior nor during the second meiotic centrosome duplication at interkinesis.
  • We have conducted a detailed description of meiotic ciliogenesis. To obtain our results, we have used known markers: SYCP3 and SYCP1 as markers of the progression of synapsis, CETN3 as a marker of meiotic centrioles, Tubulin as marker of the meiotic spindle, CEP164 as a marker of the maternal centriole, and TEX14 as a marker of cytoplasmic bridges between spermatocytes. We don’t present the description of these proteins as our own discovery in our manuscript, but rather use their known patterns of distribution to analyze the relative distribution of ciliary known markers (acTub and ARL13B). The compilation of these results allowed us to propose our model for the male mouse meiotic cilium from a histological and a subcellular perspective.

Finally, the reviewer finally points to lack of quantitative data. We agree with this comment, and we have now included different quantitative data for Figures 1, 5, 6 and 7 that confirm and add stronger confidence to the results presented. We are confident that his new version will satisfy most of the reviewer’s concerns.

We provide below a response to the specific comments made on the manuscript:

Figure 1: authors report ~20% of spermatocytes have cilia in early zygotene, no quantification is given for this claim. Spermatocytes are assigned very specific phases of meiosis (e.g. early zygotene, late zygotene) with little explanation to the classification and with no apparent markers to confirm the classification. Primary cilia in mouse spermatocytes at zygotene has been previously reported (Mytlis et al. Science (2022), Li, Xinhua et al. Cells (2021)). Authors also report the presence of acetylated tubulin on the spindle and midbody, this is already known.ç

Figure 2: authors report no primary cilia in meiosis II and the presence of acetylated tubulin on the spindle and midbody, again these are already known.

Figure 3: authors report that not all tubulin in a cell is acetylated and that αTubulin is not a reliable marker for primary cilia, these are both already known.

We thank the reviewer for these comments and are grateful to have the opportunity to respond to them.

We understand that the reviewer believes that some of the findings presented in our manuscript were already known. However, we believe that our findings are more novel than the reviewer suggests when one properly accounts for differences in the models of study, including the species used, the analysis of somatic cell cycle versus meiosis, and of oogenesis versus spermatogenesis.

Regarding the reviewer’s concerns for Figures 1-3, please find below our answers:

The quantification for the number of zygotene spermatocytes that present cilia was included in lines 128-130 of the original submission (“Quantitative analysis showed that around 20% of the spermatocytes at zygotene showed this hair-like structure (n=100 spermatocytes at zygotene per individual, three biological replicates”).  Nevertheless, we have now included a representative graph for clarification in revised Figure 1.

Spermatocytes are assigned to very specific phases of meiosis because we have used two markers that allow us to identify the exact stage of the spermatocytes in Figures 1-3: SYCP3 (synaptonemal complex protein 3) is a widely used marker of synapsis progression that is used in most, if not all, the publications focused on meiosis to identify meiocyte stages. In addition, we have also immunolocalized CETN3 to detect the moment when centrioles are duplicated, and Tubulin as a marker of the meiotic spindles. We used acTub as a marker of ciliary structures, and therefore we can conclude that the projection that we observed in spermatocytes at zygotene is indeed a cilium.

On the other hand, to our knowledge, the distribution of acTub during both mouse meiotic divisions was not previously reported. Multiple reports have described the presence of acTub in the spindle and midbody of mitotic cells, and some of them are cited in our manuscript. These publications are focused on mitotic cells, not meiocytes. We here present a new detailed pattern of distribution during the first and the second meiotic division that allow us to suggest that acTub might be contributing towards the stability of the centrioles throughout the entire meiosis, and towards the stability of the meiotic spindles I and II at metaphase I/II and the midbodies during telophase I/II. This is discussed in the section entitled “Tubulin acetylation during male mouse meiosis” of the discussion.

Figure 4: authors report that SYCP3 positive spermatocytes have primary cilia. This is reported in Mytlis et al. Science (2022) and does not add any new information about the timing of cilia formation to the data the authors have already presented, though it is a more reliable piece of data as it uses a marker to show spermatocyte’s stage of meiosis and uses an additional cilia marker.

We believe that is important to point out the exact stage where the meiotic cilium is formed in mouse spermatogenesis. Mytlis et al. Science (2022) is an outstanding publication. However, it shows a mouse meiotic cilium that barely extends the length of the basal body, and it does not identify the stage of the spermatocytes. SYCP3 positive labelling per se does not identify the stage unless a careful study of the pattern of this protein along the axial/lateral element of the synaptonemal complex is done. Our work complements the preliminary data of Mytlis et al. Science (2022) for male mouse meiosis, which is of course cited in our manuscript (see an example in line 126 of the revised version).

Figure 5: authors report that the primary meiotic cilium in mouse spermatocytes starts at 5µm in length at the beginning of zygotene, grows up to 15µm and then shrinks to 10-13µm at mid-late zygotene. No quantification is provided to support these claims. Authors should include a graph showing the measured lengths of primary cilia at early, mid, and late zygotene and show that there is a significant difference in lengths of cilia in cells at different stages in zygotene.

We are very grateful to the reviewer for suggesting this, as it has helped us improve our work. A quantitative study of the length of the cilium is now included in revised Figure 5. Differences in ciliary length at different stages are significant. It is cited in the text in revised lines 215-222 of the results section, and also included in the legend of Figure 5.

Figure 6: authors report that the mother centriole, as marked by Cep164, forms the primary cilium in mouse spermatocytes. This has already been defined as the mechanism for primary ciliogenesis (Graser et al. JCB (2007), Ishikawa & Marshall Nat Rev Mol Cell Bio (2011)).

Graser et al. JCB (2007) performed their study in somatic cells. Wellard et al (2021) show the distribution of CEP164 in mouse spermatocytes, but they do not relate it to the presence of a cilium. Our work uses the known marker of the mother centriole (CEP164) to prove that the meiotic primary cilium in mouse also emanates from the mother centriole, pointing to a conservation of the process of ciliogenesis between somatic and germ cells.

Authors also report that cilia are not present during bouquet formation during meiosis. This is contrary to the model presented by Mytlis et al. and is novel. However, the authors provide no quantification on how the presence of primary cilia in spermatocytes correlates with the presence of a bouquet, nor do they show an example of a spermatocyte with a bouquet that lacks a cilium as a control, so it is impossible to draw this conclusion from the data.

We are grateful to the reviewer for suggesting this. The quantification of the non-correlation of the presence of cilia and the bouquet configuration is now included in the text (results section lines revised Figure 6. As this data presents, 100% of the spermatocytes that present cilia are not showing a bouquet configuration (revised lines 265-271 of the results section). On the other hand, we present an image of a semi bouquet in the revised Figure 6. In this regard, we must say that the bouquet per se is extremely rarely fully observed in mouse. In fact, our squash technique allows us to visualize the cells in a 3D configuration and is common to detect most of the telomere ends in one area of the nucleus but also some of them remaining outside this semi bouquet configuration. Therefore, it is possible to observe a partial bouquet, where most, but not all, the chromosome ends are located in a particular area of the nuclear envelope. To prove this, we have included a new image in Figure 6 (Fig. 6 II C-D), where a partial bouquet conformation (labelled by TRF2) is observed in a spermatocyte at zygotene. In this cell, and all the cells at an equivalent semi bouquet arrangement, there is no presence of the meiotic cilia. If there was a direct correlation between the presence of bouquet and the meiotic cilia, we should observe the cilia in these partial bouquet conformations, especially because our results demonstrate that cilia start at the onset of synapsis (Figure 5), where no full bouquet is observed.

Figure 7: authors report that spermatocytes are connected through cytoplasmic bridges (already known). Authors also report that there is only one primary cilia per spermatocyte cyst and speculate that the function of the primary cilia is coordinating meiosis within a cyst. While this is an intriguing model the authors provide no quantification for numbers of primary cilia per cyst and provide no evidence for this mechanic action beyond an observed correlation, so it is impossible to draw this conclusion from the data.

We thank the reviewer for suggesting this and giving us the possibility to include quantification data in revised Figure 7. We believe that the observation of cysts where not all the spermatocytes are ciliated is by itself an important discovery. In addition, we have developed a quantification of the number of cysts that present only one ciliated spermatocyte. In n=25 cysts of spermatocytes at zygotene, all of them presented only one ciliated spermatocyte. This data is included in the results section in revised lines 289-294. The legend of Figure 7 II also includes this quantification data.

We believe that our work is the first to point to a potential mechanism where the cilia function of one spermatocyte might be shared between the spermatocytes of the same cyst during male mouse meiosis. Our future studies will focus on deciphering the exact signaling pathways that allow cyst progression with the presence of just one cilium.

We hope that the new data of the revised version, together with the discussion, are of interest to the reviewer. We believe that the revised version has improved our work and that the findings are of interest for the scientific community, opening the door to future deeper studies of the function and regulation of the mouse meiotic cilia in the context of spermatogenesis and ciliopathies research.

Round 2

Reviewer 2 Report

·       First, our results do not reproduce the work of previous authors. In fact, we refute one of the main statements of the report presented by Mytlis and coworkers: the ciliary structures in mouse spermatocytes are not very short, as these authors stated, but, on the contrary, cilia are strikingly large (up to 20µm). This finding was only possible by providing detailed morphological analyses that were lacking in previous studies.  We present data extracted from adult mice in a reproductive stage, rather than prepuberal 12-16 dpp mice (Mytlis et al, 2022).

This is a valid response, and a more compelling argument than the one currently presented in the text, that male mouse meiotic cilia are on average longer than meiotic cilia in zebrafish oocytes (line 496-497 in revised) especially if the bouquet data suggesting that zebrafish meiotic cilia are functionally very different from male mouse meiotic cilia is to be believed. It raises the question, are the differences reported here due solely to the different models used. For these reasons, the text should report these model differences.

  • Finally, the reviewer finally points to lack of quantitative data. We agree with this comment, and we have now included different quantitative data for Figures 1, 5, 6 and 7 that confirm and add stronger confidence to the results presented. We are confident that his new version will satisfy most of the reviewer’s concerns.

The addition of quantification to Figure 5 is appreciated, and addresses problems with that figure. The quantification given in figure 1 is inadequate as it only shows data on the phase of meiosis that the authors are interested in. A complete quantification of percent of cells with cilia during all phases of meiosis shown in the figure is required to show that the ciliation during zygotene is meaningful and unique to that phase. The inclusion of quantification within the text for figures 6 and 7 is unacceptable, graphs for this quantification must be shown in the figure itself. Quantification should be added to Figures 2, 3, and 4 as well, to support the conclusions that each figure is making.

  • On the other hand, to our knowledge, the distribution of acTub during both mouse meiotic divisions was not previously reported.  Graser et al. JCB (2007) performed their study in somatic cells. Wellard et al (2021) show the distribution of CEP164 in mouse spermatocytes, but they do not relate it to the presence of a cilium.

If there is any substantial evidence or reason to believe that these conserved processes would differ between mitosis and meiosis or between somatic and germ cells then these are indeed novel contributions. In the absence of this evidence, these pieces of data are not novel and the presentation of them as being novel is misleading.

  • In this regard, we must say that the bouquet per se is extremely rarely fully observed in mouse.

I assume this means that there is something about bouquet formation in mice that makes it difficult to observe using current techniques. If the authors are aware that it is difficult to observe a bouquet in mice organoids and were unable to collect data on cells with a bouquet, then it is disingenuous to report that cilia are not present and not functioning during bouquet formation as reported in Mytlis et al. The fact that spermatocytes are ciliated in the absence of a bouquet does not preclude the presence of a cilium in the presence of a bouquet unless the author has data to specifically show this is the case.

Author Response

Please notice that all referee´s comments are copied in blue. Author´s responses are in black.

The changes are highlighted in yellow in the revised version (word document with tracked changes)

Response to the Referee #2

     Response to R#2 First round. First, our results do not reproduce the work of previous authors. In fact, we refute one of the main statements of the report presented by Mytlis and coworkers: the ciliary structures in mouse spermatocytes are not very short, as these authors stated, but, on the contrary, cilia are strikingly large (up to 20µm). This finding was only possible by providing detailed morphological analyses that were lacking in previous studies.  We present data extracted from adult mice in a reproductive stage, rather than prepuberal 12-16 dpp mice (Mytlis et al, 2022).

Referee #2: This is a valid response, and a more compelling argument than the one currently presented in the text, that male mouse meiotic cilia are on average longer than meiotic cilia in zebrafish oocytes (line 496-497 in revised) especially if the bouquet data suggesting that zebrafish meiotic cilia are functionally very different from male mouse meiotic cilia is to be believed. It raises the question, are the differences reported here due solely to the different models used. For these reasons, the text should report these model differences.

Response to the reviewer second round: We thank the reviewer for indicating this. This additional discussion about the differences observed between both species in regard to the length of the cilium is now included in the discussion, section “The meiotic cilium in mouse spermatogenesis” (labelled in yellow in the word document with tracked changes).

Response to R#2 First round. Finally, the reviewer finally points to lack of quantitative data. We agree with this comment, and we have now included different quantitative data for Figures 1, 5, 6 and 7 that confirm and add stronger confidence to the results presented. We are confident that his new version will satisfy most of the reviewer’s concerns.

Referee #2: The addition of quantification to Figure 5 is appreciated, and addresses problems with that figure. The quantification given in figure 1 is inadequate as it only shows data on the phase of meiosis that the authors are interested in. A complete quantification of percent of cells with cilia during all phases of meiosis shown in the figure is required to show that the ciliation during zygotene is meaningful and unique to that phase. The inclusion of quantification within the text for figures 6 and 7 is unacceptable, graphs for this quantification must be shown in the figure itself. Quantification should be added to Figures 2, 3, and 4 as well, to support the conclusions that each figure is making.

Response to the reviewer second round:

Regarding Figure 1, we confirm again that the cilia are only present at zygotene. As our manuscript explains, meiotic cilia appear at the onset of synapsis and is never detected after zygotene. Therefore, we confirm that we only include the quantitative analysis of the number of ciliated spermatocytes at zygotene because it is the only stage where cilia are observed. We sincerely believe it is scientifically unnecessary to include graphs for all the stages of meiosis. We consider that the quantitative analysis of the presence of a structure in the stages where such structure is absent is not relevant, as it would indicate 0% ciliated spermatocytes in every stage except zygotene. Even so, as centrosomes duplicate twice in meiosis, and cilia appear prior to the first centrosome duplication at zygotene, we have now included a graph showing that the cilia are not detected again prior to the second centrosome duplication at interkinesis (new graph in Figure 2).

We believe that a quantitative analysis for Figures 3 and 4 are not justified. In Figure 3, representative images of the different patterns of distribution of acTubulin and non-acetylated Tubulin are presented. All stages at the equivalent stage present the same distribution. In Figure 4, we are presenting representative histological sections, and representative images of the colocalization of acTub and ARL13B are shown. Therefore, a quantitative analysis is unnecessary as the figures present images which are representative of the distribution of these proteins at the indicated stages of meiosis.

In Figure 5, we have now included the pachytene stage in the graph, where, as it was already mentioned in the text, the cilia are not present and thus the length of the structure is not measurable. To assure this, we have quantified n=100 pachytenes, corroborating that none of them were ciliated.

Quantification representative graphs for Figures 6 and 7 are now included in the figures, and also cited in the text and legends upon referee´s suggestion.

Response to R#2 First round. On the other hand, to our knowledge, the distribution of acTub during both mouse meiotic divisions was not previously reported. Graser et al. JCB (2007) performed their study in somatic cells. Wellard et al (2021) show the distribution of CEP164 in mouse spermatocytes, but they do not relate it to the presence of a cilium.

Referee #2: If there is any substantial evidence or reason to believe that these conserved processes would differ between mitosis and meiosis or between somatic and germ cells then these are indeed novel contributions. In the absence of this evidence, these pieces of data are not novel and the presentation of them as being novel is misleading.

Response to the reviewer second round. We do not fully understand this comment. Mitosis and meiosis are strikingly different processes. Particularly, the events occurring during first meiotic prophase (chromosome pairing, recombination) do not have any comparable counterpart in mitosis. Meiosis is a much longer process, lasting about 10 days in the mouse, compared to about one hour for mitosis.  In addition, meiosis comprises two centrosome duplication and therefore the formation of two consecutive bipolar spindles without a DNA replication event in between. This means than many morphological and physiological events that occur during meiosis are inconceivable during mitosis. Therefore, contrary to reviewer’s opinion, we suggest that there is indeed no reason to think that a particular cellular event, as ciliogenesis or its related processes, should behave in meiosis in a similar way it does during mitosis. The same argument can be applied to the differences on the formation of the cilium in any of the different somatic cell types of a mammalian organism. The formation of the cilium in lung cells does not allow us to make any substantial prediction about its dynamics in nerve, kidney or germ cells. This being said, literature is plenty of publications whose main aim is the comparison of mitotic versus the meiotic cellular events. Many of these reports have highlighted that somatic cell cycle and meiosis are indeed two very different processes, hence the events that occur in both types of cell divisions are sometimes comparable, but others are not.                       

For these reasons, we believe that addressing the distribution of acTub in both meiotic spindles is relevant, and this has allowed us to discuss the function of this posttranslational modification of Tubulin in the section of the discussion “Tubulin acetylation during male mouse meiosis”. Regarding CEP164, we find relevant concluding that miotic cilia emanate from the maternal centrosomes prior to the first centrosome duplication at zygotene, whereas a new ciliogenesis from the new distal appendages does not occur in the second centrosome duplication at interkinesis. To reinforce this conclusion, a new sentence has been added to the discussion section “Meiotic centrosomes and their relation to ciliogenesis and flagelogenesis” (labelled in yellow in the revised version). We therefore think that our report contributes to adding evidence to the differences in cilia formation in meiotic versus mitotic cells.

Response to R#2 First round. In this regard, we must say that the bouquet per se is extremely rarely fully observed in mouse.

Referee #2: I assume this means that there is something about bouquet formation in mice that makes it difficult to observe using current techniques. If the authors are aware that it is difficult to observe a bouquet in mice organoids and were unable to collect data on cells with a bouquet, then it is disingenuous to report that cilia are not present and not functioning during bouquet formation as reported in Mytlis et al. The fact that spermatocytes are ciliated in the absence of a bouquet does not preclude the presence of a cilium in the presence of a bouquet unless the author has data to specifically show this is the case.

Response to the reviewer second round: In our research group we have collected data about the bouquet in several mammalian species, as for example in marsupials (1). However, this structure significantly differs between species. Since its discovery in mouse, the bouquet conformation has been demonstrated as an extremely transient stage, as repeatedly reported by different experts of the field. Telomere clustering occurs in a time window limited to the onset of zygotene, which represent ∼0.5% of spermatocytes, and for a very short period of time (2). Moreover, in those cells the bouquet does not usually comprise all the chromosomal ends, behaving in fact as a semi-bouquet (2,3). In our experiment, we were able to find these zygotene spermatocytes with a partial confluence of chromosomal ends in a semi-bouquet, but the full bouquet configuration is extremely rare and certainly would extend for less time than the cilia (from the onset of synapsis until mid-zygotene). This is clearly different to other species like zebrafish, where the bouquet configuration is obvious and easily detected (4,5); or C. elegans, where the bouquet does not exist per se but the chromatin concentrates at one side of the nucleus and adopts a characteristic half-moon shape (6); or as we have mentioned above, the conspicuous bouquet in marsupials (1).

Therefore, different organisms present distinct chromosome configurations during prophase I. We believe that Mytlis et al data for zebrafish are excellent, but this publication does not address the potential relationship between mouse meiotic cilia and the presence or absence or bouquet, either in mouse oocytes (Figure 6 and Supplementary Figure 18, Myltlis et al 2022) or spermatocytes (Supplementary Fig 19, Mytlis et al 2022). None of the images presented in that report showed labelling for telomeric ends. Therefore, we believe that zebrafish and mouse behave different, and thus our work is adequate in discussing this topic. For this reason, we included a new image of a representative semi-mouse bouquet spermatocyte at zygotene with TRF2 labelling (Fig. 6 II.C,D). In all the equivalent spermatocytes, as the new graph shows (Fig 6 III), there is no presence of the meiotic cilia. With these data, we reinforce our suggestion that there is no direct relation nor concurrency between the representative mouse semi-bouquet and the presence of meiotic cilia.

Additional discussion about the differences observed between the bouquet conformation of telomeric ends between different species is now included in the discussion section (section Zygotene cilia and bouquet formation are not directly related in male mouse meiosis).

  1. Page, J. et al., The pairing of X and Y chromosomes during meiotic prophase in the marsupial species Thylamys elegans is maintained by a dense plate developed from their axial elements. J. Cell Sci. 2003, 116, p551-560.
  2. Liebe, B., et al., Mutations that affect meiosis in male mice influence the dynamics of the mid-preleptotene and bouquet stages. Exp Cell Res, 2006. 312(19): p. 3768-81.
  3. Enguita-Marruedo, A., et al., Transition from a meiotic to a somatic-like DNA damage response during the pachytene stage in mouse PLoS Genet, 2019. 15(1): p. e1007439.
  4. Siegfried, K.R., It starts at the ends: The zebrafish meiotic bouquet is where it all begins. PLoS Genet, 2019. 15(1): p. e1007854.
  5. Imai, Y., et al., Meiotic Chromosome Dynamics in Zebrafish. Front Cell Dev Biol, 2021. 9: p. 757445.
  6. Link, J. and V. Jantsch, Meiotic chromosomes in motion: a perspective from Mus musculus and Caenorhabditis elegans. Chromosoma, 2019. 128(3): p. 317-330.
